# CONDA: Adaptive Concept Bottleneck for Foundation Models Under Distribution Shifts

**Jihye Choi, Jayaram Raghuram, Yixuan Li & Somesh Jha**
Department of Computer Sciences, University of Wisconsin - Madison
{jihye,jayaramr,sharonli,jha}@cs.wisc.edu

## Abstract

Advancements in foundation models (FMs) have led to a paradigm shift in machine learning. The rich, expressive feature representations from these pre-trained, large-scale FMs are leveraged for multiple downstream tasks, usually via lightweight fine-tuning of a shallow fully-connected network following the representation. However, the non-interpretable, black-box nature of this prediction pipeline can be a challenge, especially in critical domains, such as healthcare, finance, and security. In this paper, we explore the potential of Concept Bottleneck Models (CBMs) for transforming complex, non-interpretable foundation models into interpretable decision-making pipelines using high-level concept vectors. Specifically, we focus on the test-time deployment of such an interpretable CBM pipeline "in the wild", where the distribution of inputs often shifts from the original training distribution. We first identify the potential failure modes of such pipelines under different types of distribution shifts. Then we propose an *adaptive concept bottleneck* framework to address these failure modes, that dynamically adapts the concept-vector bank and the prediction layer based solely on unlabeled data from the target domain, without access to the source dataset. Empirical evaluations with various real-world distribution shifts show our framework produces concept-based interpretations better aligned with the test data and boosts post-deployment accuracy by up to 28%, aligning CBM performance with that of non-interpretable classification [1].

## 1 Introduction

Foundation Models (FMs), trained on vast data, are powerful feature extractors applicable across diverse distributions and downstream tasks (Bommasani et al., 2021; Rombach et al., 2022). They can be applied to classification tasks off-the-shelf via zero-shot prediction, or via linear probing using task-specific fine-tuning data (Kumar et al., 2022; Radford et al., 2021). Despite these strong advantages, foundation model-based systems often operate as inscrutable black-boxes, presenting a barrier to user trust and wider deployment in safety-critical settings. Another challenge faced in the standard deployment of FM-based deep classifiers is their vulnerability to distribution shifts at test time caused *e.g.*, due to environmental changes, which can cause a drop in performance (Bommasani et al., 2021). This is particularly challenging in high-stakes domains such as healthcare (AlBadawy et al., 2018; Eslami et al., 2023), autonomous driving (Yu et al., 2020), and finance (Wu et al., 2023a).

In this work, we address these challenges by developing an *interpretable classification* framework that enjoys the rich, expressive feature representations of FMs, while also having enhanced robustness towards *distribution shifts at test time*. To tackle interpretability, we utilize *Concept Bottleneck Models (CBMs)* (Koh et al., 2020), transforming FM-based classifiers into interpretable, concept-based prediction pipelines. With the rapid advancements in FMs, there is strong opportunity to utilize them as powerful backbones, providing robust feature representations from which high-quality concepts can be extracted. Unlike early CBM approaches that required expensive concept annotations, recent advances show potential for constructing concept bottlenecks without any annotations by leveraging vision-language models (Oikarinen et al., 2023; Wu et al., 2023b), and achieving performance on par with non-interpretable models. Concept-based predictions provide not only interpretability, but are

---

[1]The code repository for our work is available at `https://github.com/jihyechoi77/CONDA`.

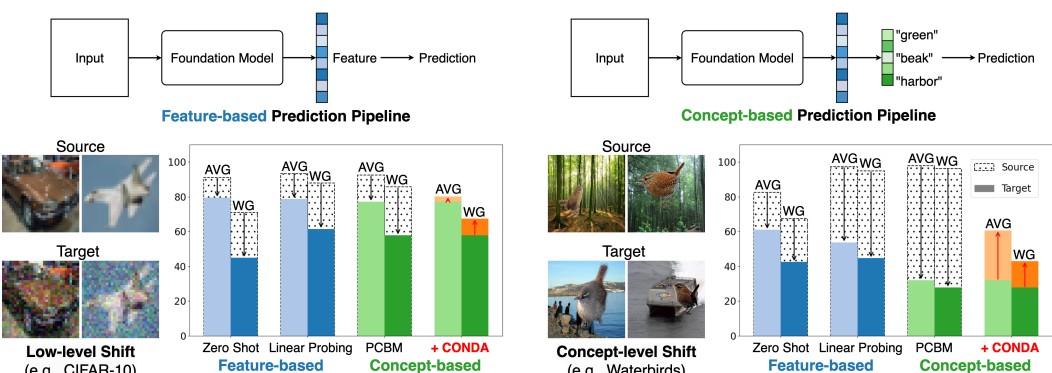

Figure 1: **Concept-based predictions are not inherently more robust to distribution shifts than feature-based predictions, necessitating dynamic adaptation after deployment.** We observe significant drops in the averaged group accuracy (AVG) and worst-group accuracy (WG) from the source to the target (test) domain under two types of distribution shifts: (1) *low-level shift* (left), where inputs are perturbed without modifying class-level semantics (*e.g.*, Gaussian noise); and (2) *concept-level shift* (right), where some high-level semantics change. On the left, predictions made through high-level concepts (*e.g.*, by PCBM (Yuksekgonul et al., 2023) here) are not necessarily more robust to low-level input perturbations. On the right, the performance of concept-based predictions suffers an even more drastic drop, failing to leverage the expressiveness of the foundation model's high-level features, and falling behind direct feature-based predictions (here zero-shot and linear-probing based classification). However, with CONDA, we can boost the performance of the deployed concept-based predictor to be on par with, or even better than, its non-interpretable counterparts.

also beneficial for *robustness*; a central premise of CBMs is that as complex feature embeddings go through the concept bottleneck, the resulting predictions should, in theory, become more invariant to inconsequential input changes (Kim et al., 2018; Adebayo et al., 2020).

However, we observe that CBMs directly deployed under distribution shifts often do not produce more robust predictions compared to FM-based classifiers (either in zero-shot or fine-tuned configurations). For instance, as illustrated in Figure 1, even when a concept-based prediction pipeline matches or outperforms a feature-based prediction pipeline in the training (source) domain, its test-time (deployment) performance can drop significantly under distribution shifts. This highlights that a naive adoption of CBMs is insufficient for fully leveraging the robustness and expressiveness of FM features under test-time shifts, necessitating a dynamic approach for adapting concept-based predictions in real-world deployments.

The problem of test-time (or source-free domain) adaptation (TTA) has recently been explored extensively (Wang et al., 2021; Jung et al., 2023; Liang et al., 2023). The goal of TTA is to adapt a deep classifier, trained on source domain data, to a test-time deployment setting where there could be distribution shifts (*e.g.*, corruptions, environment changes), and given access to *only* unlabeled test data and the source domain classifier. While the main focus of TTA methods has been on non-interpretable, deep classifier networks, we present the first approach (to our knowledge) for *TTA of concept bottlenecks with a foundation model backbone*. Our key contributions are summarized as follows: given unlabeled test data, a frozen FM, and a pre-constructed concept bottleneck, we

1. formally categorize the types of distribution shifts expected during deployment, identifying possible failure modes of the concept bottleneck pipeline under these shifts (Section 2);

2. propose a novel framework CONDA (CONcept-based Dynamic Adaptation), where each component of the framework is adapted based on the identified failure modes, without requiring access to the source dataset or any labels for the test data (Section 3);

3. empirically demonstrate the robustness and interpretability of CONDA across various FM backbones (*e.g.*, CLIP:ViT-L/14) and concept bottleneck construction methods (*e.g.*, post-hoc CBM), showing that CONDA improves the test-time accuracy by up to 28%, and provides concept-based interpretations better tailored towards the test inputs (Section 4).

**Related Work.** Distribution shifts occur when the data distribution during deployment differs from that during training, leading to degraded model performance (Quiñonero-Candela et al., 2022). To address this issue, TTA methods adapt the model parameters using *only* unlabeled test data; via

entropy minimization (Wang et al., 2021; Zhang et al., 2022), self-supervised learning at test time (Sun et al., 2020), class-aware feature alignment (Jung et al., 2023), and updating batch normalization statistics using test data (Nado et al., 2020). These methods enable models to adapt on-the-fly without requiring access to the labeled training data. In the era of foundation models, recent efforts have been made to enhance their zero-shot inference robustness under distribution shifts without modifying their internal parameters (Chuang et al., 2023; Adila et al., 2024). However, improving the robustness of the foundation model itself is not the focus of our work. Instead, given *any* foundation model, regardless of its inherent robustness, we aim to construct an interpretable framework without sacrificing the utility, striving for performance that matches or exceeds that of the foundation model's feature-based predictions. Refer to Appendix A for extended related work.

## 2 CONCEPT BOTTLENECK MODEL UNDER DISTRIBUTION SHIFTS

### 2.1 BACKGROUND: FOUNDATION MODELS WITH A CONCEPT BOTTLENECK

Consider a foundation model $\phi : \mathcal{X} \mapsto \mathbb{R}^d$, which is any pre-trained backbone model or feature extractor (Eslami et al., 2023; Jia et al., 2021; Girdhar et al., 2023) that maps the input $\mathbf{x}$ to an intermediate feature embedding $\phi(\mathbf{x}) \in \mathbb{R}^d$. $\phi(\mathbf{x})$ is pre-trained on a large-scale, broad mixture of data for general purposes, *i.e.*, not restricted to a specific domain. For a specific downstream classification task, the general practice is to either apply zero-shot prediction on $\phi(\mathbf{x})$, or to train a shallow label predictor $\mathbf{g}_s : \mathbb{R}^d \mapsto \mathbb{R}^L$, that maps $\phi(\mathbf{x})$ to the un-normalized class predictions $\mathbf{g}_s(\phi(\mathbf{x}))$, using a supervised loss (*e.g.*, cross-entropy).

A CBM (Koh et al., 2020) first projects the high-dimensional feature embedding to a lower $m$-dimensional ($m \ll d$) *concept-score space* (acting like a bottleneck), and follows it with a *label predictor*, which is a simple affine or fully-connected layer that maps the concept scores into class predictions. The concept bottleneck is represented by a matrix of $m$ unit-norm *concept vectors* $\mathbf{C}_s = [\mathbf{c}_{s1} / \|\mathbf{c}_{s1}\|_2 \ \cdots \ \mathbf{c}_{sm} / \|\mathbf{c}_{sm}\|_2]^\top \in \mathbb{R}^{m \times d}$, where each $\mathbf{c}_{si} \in \mathbb{R}^d$ represents a high-level concept (*e.g.*, "stripes", "fin", "dots"). The $m$ concept scores are obtained via a linear projection $\mathbf{v}_{\mathbf{C}_s}(\mathbf{x}) = \mathbf{C}_s \, \phi(\mathbf{x})$, which is followed by a fully-connected layer to obtain the CBM model as

$$\mathbf{f}_s^{(\text{cbm})}(\mathbf{x}) := \mathbf{W}_s \, \mathbf{v}_{\mathbf{C}_s}(\mathbf{x}) + \mathbf{b}_s = \mathbf{W}_s \mathbf{C}_s \, \phi(\mathbf{x}) + \mathbf{b}_s = \mathbf{g}_s(\phi(\mathbf{x})) \tag{1}$$

The label predictor $\mathbf{g}_s(\mathbf{z})$ is defined by the parameters $\mathbf{W}_s \in \mathbb{R}^{L \times m}$, $\mathbf{b}_s \in \mathbb{R}^L$, and $\mathbf{C}_s$. A key advantage of the CBM is that its predictions are an affine combination of the high-level concept scores, which allows for better interpretability of the model. Since the label predictor of a CBM is chosen to be simple, its performance is strongly dependent on the construction of the concept bank. Additional details on the preparation of concept vectors can be found in Appendix A.

### 2.2 DISTRIBUTION SHIFTS IN THE WILD

We define the *source domain* $D_s$ and the *target domain* $D_t$. Let $\mathcal{H}$ be a *concept hypothesis* class, defined as the space of measurable concept mappings $\mathbf{h} : \mathbb{R}^d \to \mathbb{R}^m$ from the feature representation $\phi(\mathbf{x})$ to concept scores. We also define the *concept set* $\mathcal{C} := \{c_1, c_2, \cdots, c_m\}$, where each $c_i : \mathbb{R}^d \mapsto \mathbb{R}$ represents a high-level concept mapping (*e.g.*, stripe pattern, grass, beach, *etc.*). For a domain $D_j$, $j \in \{s, t\}$, we define the concept score distribution as $\mathbb{P}_{\text{con}}(D_j, \phi, \mathbf{h}) = (\mathbf{h} \circ \phi)_* \mathbb{P}_j$, where $(\mathbf{h} \circ \phi)_* \mathbb{P}_j$ is the push-forward measure (Le Gall, 2022) of $\mathbb{P}_j$ under $\mathbf{h} \circ \phi$. Note that $\mathbf{h}$ is determined by $\mathcal{C}$ such that $\mathbf{h}(\phi(\mathbf{x})) = [c_1(\phi(\mathbf{x})), \cdots, c_m(\phi(\mathbf{x}))]^T$ [2].

Let $\mathcal{G}$ be a *classification hypothesis* class, defined as a set of measurable classifiers $\mathbf{g} : \mathbb{R}^m \to \mathbb{R}^L$ mapping the concept scores to prediction logits. Finally, we define the distribution of predictions as the push-forward measure of $\mathbb{P}_{\text{con}}(D_j, \phi, \mathbf{h})$ under $\mathbf{g}$: $\mathbb{P}_{\text{pred}}(D_j, \phi, \mathbf{h}, \mathbf{g}) = \mathbf{g}_* \mathbb{P}_{\text{con}}(D_j, \phi, \mathbf{h})$.

Given $\mathbf{h} \in \mathcal{H}$ and $\mathbf{g} \in \mathcal{G}$, we categorize the distribution shifts in the target domain, $\{\mu_t(\mathbf{t}_i) > 0 \mid \mathbf{t}_i \in \mathcal{T}\}$, into one of the following broad categories:

1. **Low-level shift:** This type of transformation does not change the concept score distribution across the domains. Examples include additive Gaussian noise, blurring, and pixelization, which employ

---

[2] A common approach is to define $c_i(\phi(\mathbf{x}))$ as the inner product of a (unit-normalized) concept vector with the feature representation $\phi(\mathbf{x})$, which results in a score for concept $i$.

low-level changes to the input (*e.g.*, CIFAR10-C (Hendrycks & Dietterich, 2019)):

$$\mathbb{P}_{\text{con}}(D_t, \phi, \mathbf{h}) = \mathbb{P}_{\text{con}}(D_s, \phi, \mathbf{h}) \tag{2}$$

Naturally, the resulting distribution of predictions based on the concept scores also remains the same across the domains, *i.e.*, $\mathbb{P}_{\text{pred}}(D_s, \phi, \mathbf{h}, \mathbf{g}) = \mathbb{P}_{\text{pred}}(D_t, \phi, \mathbf{h}, \mathbf{g})$.

2. **Concept-level shift:** This type of transformation alters the concept score distribution, but not the prediction distribution across the domains. Examples include replacing water background with a land background in images (*e.g.*, Waterbirds, Metashift (Sagawa et al., 2019; Liang & Zou, 2021)):

$$\mathbb{P}_{\text{con}}(D_t, \phi, \mathbf{h}) \neq \mathbb{P}_{\text{con}}(D_s, \phi, \mathbf{h})$$
$$\mathbb{P}_{\text{pred}}(D_t, \phi, \mathbf{h}, \mathbf{g}) = \mathbb{P}_{\text{pred}}(D_s, \phi, \mathbf{h}, \mathbf{g}) \tag{3}$$

**Definition 1** *The concept set $\mathcal{C} = \{c_1, c_2, \ldots, c_m\}$ is **complete** if there exists a classifier $\mathbf{g} \in \mathcal{G}$ such that, for both low-level and concept-level shifts, the prediction distributions conditioned on the concepts are identical for source and target domains:*

$$\mathbb{P}_{pred}(D_s, \phi, \mathbf{h}, \mathbf{g}) = \mathbb{P}_{pred}(D_t, \phi, \mathbf{h}, \mathbf{g}). \tag{4}$$

*This implies that there exists a mapping from concept scores to labels encompassing both the source and target domains.*

### 2.3 Failure Modes of Concept Bottleneck for Foundation Models

Based on the definitions above, we categorize the possible failure modes of the decision-making pipeline of a foundation model equipped with a CBM, defined by a given $D_s, D_t, \phi, \mathbf{h} \circ \phi = [c_1 \circ \phi, \cdots, c_m \circ \phi]$, and $\mathbf{g}$ as follows.

1. **Non-robust concept bottleneck under low-level shift:** the concept mapping $\mathbf{h}$ is *not* robust to low-level shifts, causing discrepancies in the concept-level predictions:

$$\mathbb{P}_{\text{con}}(D_t, \phi, \mathbf{h}) \neq \mathbb{P}_{\text{con}}(D_s, \phi, \mathbf{h}),$$

violating the requirement for a low-level shift in Eqn. 2. Such discrepancies in the concept predictions can lead to degraded performance in $D_t$, resulting from mismatched prediction distributions, *i.e.*, $\mathbb{P}_{\text{pred}}(D_t, \phi, \mathbf{h}, \mathbf{g}) \neq \mathbb{P}_{\text{pred}}(D_s, \phi, \mathbf{h}, \mathbf{g})$.

2. **Non-robust classifier under concept-level shift:** Given that the concept score distributions differ due to a concept-level shift as in Eqn. 3, the given classifier $\mathbf{g}$ fails to produce consistent prediction distributions across the domains, violating Eqn 3:

$$\mathbb{P}_{\text{pred}}(D_t, \phi, \mathbf{h}, \mathbf{g}) \neq \mathbb{P}_{\text{pred}}(D_s, \phi, \mathbf{h}, \mathbf{g})$$

3. **Incomplete concept set:** The concept set $\{c_1, c_2, \ldots, c_m\}$ is *not* complete, and there does not exist *any* $\mathbf{g} \in \mathcal{G}$ such that $\mathbb{P}_{\text{pred}}(D_s, \phi, \mathbf{h}, \mathbf{g}) = \mathbb{P}_{\text{pred}}(D_t, \phi, \mathbf{h}, \mathbf{g})$. Intuitively, it fails to capture all the necessary information for consistent predictions across domains, and Definition 1 is not achievable in the first place.

## 3 CONDA: Concept-based Dynamic Adaptation

To address the failure modes of a CBM identified in the previous section, here we propose a dynamic approach for adaptation of a CBM based *only* on unlabeled test data. We follow the setting of test-time adaptation, where the foundation model $\phi(\mathbf{x})$ and CBM, consisting of the concept bank $\mathbf{C}_s$ and label predictor $(\mathbf{W}_s, \mathbf{b}_s)$, trained on the source domain are given (see Eqn 1), but the source (training) dataset is not available. Let $\mathcal{D}_t = \{\mathbf{x}_{tn}\}_{n=1}^{N_t}$ be the unlabeled test set from the target distribution. To address the three potential failure modes in a CBM pipeline identified in Section 2.3, we propose the following three-step adaptation procedure, with each step designed to tackle a specific failure mode:

1. **Concept-Score Alignment (CSA):** The goal of this step is to perform a feature alignment of the concept scores of test inputs $\mathbf{v}_\mathbf{C}(\mathbf{x}_t) \in \mathbb{R}^m$ such that their class-conditional distributions are close to that of the concept scores in the source dataset [3]. By adapting the concept vectors $\mathbf{C}$, this will ensure that the label predictor continues to "see" very similar class-conditional input distributions at test time, thereby maintaining accurate predictions.

---

[3]We drop the subscript 's' to denote that they are adaptation parameters, not specific to the source domain.

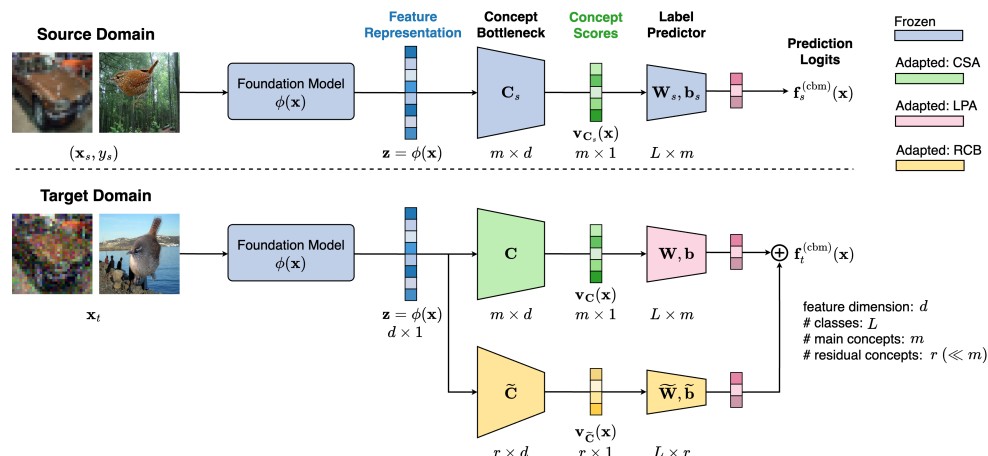

Figure 2: **Overview of CONDA, our proposed adaptation framework.** The foundation model and CBM pipeline trained on the source domain is shown at the top, while the adapted CBM, consisting of a main branch and residual branch, is shown at the bottom. The components that are adapted during each stage of the proposed method (*i.e.*, CSA, LPA, and RCB) are shown in different colors.

2. **Linear Probing Adaptation (LPA):** To further address any discrepancy or mismatch in the feature alignment CSA step (*e.g.*, due to distribution assumptions), here we adapt the label predictor $(\mathbf{W}, \mathbf{b})$ of the CBM, with the concept vectors fixed at their updated values from the CSA step.

3. **Residual Concept Bottleneck (RCB):** As discussed in Section 2.3, the concept bank from the source domain could be incomplete, and new concepts may be required to bridge the distribution gap between the domains. In this step, we introduce a residual CBM with additional concept vectors and a linear predictor, which are jointly optimized (with the parameters of the main CBM fixed) to improve the test accuracy.

**Target Domain CBM.** Figure 2 shows the overall architecture of CONDA. The residual concept bottleneck is shown as a separate branch, where we introduce $r$ additional concept vectors $\widetilde{\mathbf{C}} = [\, \widetilde{\mathbf{c}}_1 \, / \, \|\widetilde{\mathbf{c}}_1\|_2 \; \cdots \; \widetilde{\mathbf{c}}_r \, / \, \|\widetilde{\mathbf{c}}_r\|_2 \,]^\top \in \mathbb{R}^{r \times d}$. The concept scores are obtained by projecting the feature representation $\phi(\mathbf{x})$ on these residual concept vectors, and the scores are passed to another linear predictor $(\widetilde{\mathbf{W}}, \widetilde{\mathbf{b}})$ to obtain the un-normalized class predictions (logits) of the residual CBM: $\widetilde{\mathbf{W}}\widetilde{\mathbf{C}}\,\phi(\mathbf{x}) + \widetilde{\mathbf{b}}$. The un-normalized predictions of the target domain CBM are obtained by adding that of the main and the residual branch CBMs, giving

$$
\begin{aligned}
\mathbf{f}_t^{(\mathrm{cbm})}(\mathbf{x}) &= \mathbf{WC}\,\phi(\mathbf{x}) + \mathbf{b} + \widetilde{\mathbf{W}}\widetilde{\mathbf{C}}\,\phi(\mathbf{x}) + \widetilde{\mathbf{b}} \\
&= (\mathbf{WC} + \widetilde{\mathbf{W}}\widetilde{\mathbf{C}})\,\phi(\mathbf{x}) + \mathbf{b} + \widetilde{\mathbf{b}} = \mathbf{W}_{\mathrm{con}}\mathbf{C}_{\mathrm{con}}\,\phi(\mathbf{x}) + \mathbf{b}_{\mathrm{con}},
\end{aligned}
\tag{5}
$$

where $\widetilde{\mathbf{W}} \in \mathbb{R}^{L \times r}$ and $\widetilde{\mathbf{b}} \in \mathbb{R}^L$. For comparison with the source domain CBM (Eqn. 1), we have defined the combined parameters from the main and residual branch CBMs as $\mathbf{W}_{\mathrm{con}} = [\mathbf{W}\ \widetilde{\mathbf{W}}] \in \mathbb{R}^{L \times (m+r)}$, $\mathbf{C}_{\mathrm{con}} = [\mathbf{C}\,;\widetilde{\mathbf{C}}] \in \mathbb{R}^{(m+r) \times d}$, and $\mathbf{b}_{\mathrm{con}} = \mathbf{b} + \widetilde{\mathbf{b}} \in \mathbb{R}^L$. That is, adding the residual CBM is equivalent to introducing $r$ additional rows (columns) in the concept (weight) matrix. For adaptation, the parameters of the main CBM $\{\mathbf{C}, \mathbf{W}, \mathbf{b}\}$ are initialized to their corresponding values from the source domain, while the parameters of the residual CBM $\{\widetilde{\mathbf{C}}, \widetilde{\mathbf{W}}, \widetilde{\mathbf{b}}\}$ are initialized randomly.

**Pseudo-labeling.** Since the test samples are unlabeled, it becomes challenging to design adaptation objectives that can minimize a smooth proxy of the classification error rate on the target distribution. We utilize the idea of pseudo-labeling to address this, as commonly done in the TTA and semi-supervised learning literature (Chen et al., 2022; Lee et al., 2013; Sohn et al., 2020). We leverage the fact that the feature extraction backbone $\phi(\mathbf{x})$ is a foundation model that is pre-trained on diverse data distributions, and as a result is likely to be relatively robust to distribution shifts. We take an ensemble of the commonly used *zero-shot* predictor (as done *e.g.*, in Radford et al. (2021)) and a *linear probing* predictor (trained on the source dataset on top of the foundation model) to get the pseudo-labels for test samples. We combine the two by taking the class predicted with higher confidence across

both predictors. We note that more advanced pseudo-labeling methods *e.g.*, involving weak- and strong-augmentations, and soft nearest-neighbor voting (Chen et al., 2022) can be used to potentially improve our method (see Appendix D for further exploration).

Following the convention in the TTA literature (Wang et al., 2021; Chen et al., 2022), we randomly split the test data into fixed-size batches $\mathcal{D}_t = \bigcup_{b=1}^{B} \mathcal{D}_t^b$, and perform adaptation sequentially on each batch $b$, obtaining the adapted model's predictions on the same batch, before moving to the next one. Also, the CBM parameters (main and residual) are adapted in an online fashion (not episodically) (Wang et al., 2021), *i.e.*, the adapted parameters learned from a batch are used to initialize the next batch and so on [4]. For convenience, we define the test dataset with paired pseudo-labels as $\widehat{\mathcal{D}}_t = \{(\mathbf{x}_{tn}, \widehat{y}_{tn})\}_{n=1}^{N_t}$, and the corresponding pseudo-labeled test batches as $\widehat{\mathcal{D}}_t^b$, $b \in [B]$. We next expand on each stage of the CBM adaptation outlined earlier, and provide a complete algorithm for the same in Algorithm 1 in the Appendix.

## 3.1 CONCEPT SCORE ALIGNMENT

From Figure 2 (top half) and Eqn. 1, the concept scores $\mathbf{v}_{\mathbf{C}_s}(\mathbf{x}) \in \mathbb{R}^m$ are input to the linear label predictor $\mathbf{W}_s \mathbf{v} + \mathbf{b}_s$. Let $\{\mathbb{P}(\mathbf{v}_{\mathbf{C}_s}(\mathbf{x}_s) \mid y_s = y), \ y \in \mathcal{Y}\}$ be the class-conditional distributions of these concept scores on the source domain. At test time, if the distribution of the input changes such that $\mathbf{x}_t \sim p_t(\mathbf{x})$, then there is a corresponding change in the class-conditional distributions of concept scores $\{\mathbb{P}(\mathbf{v}_{\mathbf{C}_s}(\mathbf{x}_t) \mid y_t = y) = \mathbb{P}(\mathbf{C}_s \phi(\mathbf{x}_t) \mid y_t = y), \ y \in \mathcal{Y}\}$. The goal of concept-score alignment (CSA) is to adapt the source domain concept bank $\mathbf{C}_s$ to a target domain-specific one $\mathbf{C}_t$ such that the class-conditional distributions after adaptation are close to that of the source domain under some distributional distance (*e.g.*, Kullback-Leibler or Total-variation). Informally, we wish to find an adapted concept bank $\mathbf{C}_t$, starting from $\mathbf{C}_s$, such that

$$\mathbb{P}(\mathbf{C}_t \phi(\mathbf{x}_t) \mid y_t = y) \ \approx \ \mathbb{P}(\mathbf{C}_s \phi(\mathbf{x}_s) \mid y_s = y), \ \ \forall y \in \mathcal{Y}.$$

If the class priors $\{\mathbb{P}(y_t = y), \ \forall y\}$ do not change significantly, this can ensure that the label predictor of the main CBM continues to receive concept scores from a similar distribution as the source domain.

We model the class-conditional distributions of the concept scores in the source domain as multivariate Gaussians: $\mathbb{P}(\mathbf{v}_{\mathbf{C}_s}(\mathbf{x}_s) \mid y_s = y) = \mathcal{N}(\mathbf{v}_{\mathbf{C}_s}(\mathbf{x}_s); \boldsymbol{\mu}_y, \boldsymbol{\Sigma}_y), \ \forall y \in \mathcal{Y}$. Given a labeled source-domain dataset, it is straight-forward to estimate $\boldsymbol{\mu}_y$ and $\boldsymbol{\Sigma}_y$ using the sample mean and sample covariance of $\mathbf{v}_{\mathbf{C}_s}(\mathbf{x}_s)$ on the data subset from class $y$ (max-likelihood estimate). Although we cannot access the source domain dataset during adaptation, we assume to have access to these distribution statistics $\{(\boldsymbol{\mu}_y, \boldsymbol{\Sigma}_y)\}_{y \in \mathcal{Y}}$. At test time, changes to the distribution of the concept scores can be captured by a concept matrix $\mathbf{C}$ (to be adapted). For a test input $\mathbf{x}_t$, the distance of its concept scores $\mathbf{v}_{\mathbf{C}}(\mathbf{x}_t)$ from the Gaussian distribution of class $y$ is given by the *Mahalanobis metric* $D_{\text{mah}}(\mathbf{x}_t; \boldsymbol{\mu}_y, \boldsymbol{\Sigma}_y) = (\mathbf{v}_{\mathbf{C}}(\mathbf{x}_t) - \boldsymbol{\mu}_y)^\top \boldsymbol{\Sigma}_y^{-1} (\mathbf{v}_{\mathbf{C}}(\mathbf{x}_t)) - \boldsymbol{\mu}_y)$.

**Intra-class and Inter-class Distances.** Taking the pseudo-label $\widehat{y}_t$ as a proxy for the true label of $\mathbf{x}_t$, the *intra-class (or within-class)* distance measures the closeness of $\mathbf{x}_t$ to samples from its own class, while the *inter-class (or between-class)* distance measures the separation of $\mathbf{x}_t$ to samples from the other classes. They are defined as follows:

$$D_{\text{intra}}(\mathbf{x}_t, \widehat{y}_t) \ = \ D_{\text{mah}}(\mathbf{x}_t; \boldsymbol{\mu}_{\widehat{y}_t}, \boldsymbol{\Sigma}_{\widehat{y}_t}) \ \ \text{and} \tag{6}$$

$$D_{\text{inter}}(\mathbf{x}_t, \widehat{y}_t) \ = \ \frac{1}{L-1} \sum_{\ell=1:\ell \neq \widehat{y}_t}^{L} D_{\text{mah}}(\mathbf{x}_t; \boldsymbol{\mu}_\ell, \boldsymbol{\Sigma}_\ell). \tag{7}$$

Motivated by class-aware feature alignment CAFA (Jung et al., 2023), we explore an adaptation loss $\ell_{ada}$ that is specifically designed to achieve concept-score alignment on a per-class level. This loss is based on the idea that for discriminative feature alignment, the intra-class distances should be small and the inter-class distances should be large on the test samples (Ye et al., 2021; Ming et al., 2023).

$$\ell_{ada}(\mathbf{v}_{\mathbf{C}}(\mathbf{x}_t), \widehat{y}_t) \ = \ \log \frac{D_{\text{intra}}(\mathbf{x}_t, \widehat{y}_t)}{D_{\text{inter}}(\mathbf{x}_t, \widehat{y}_t)}. \tag{8}$$

---

[4] In the episodic approach, parameters would be reset to their source domain values to initialize each batch.

With this setup, we propose the adaptation objective for CSA to minimize on a test batch:

$$L_{\text{CSA}}(\mathbf{C}) = \frac{1}{|\widehat{\mathcal{D}}_t^b|} \sum_{(\mathbf{x}_t, \widehat{y}_t) \in \widehat{\mathcal{D}}_t^b} \ell_{ada}(\mathbf{v_C}(\mathbf{x}_t), \widehat{y}_t) + \lambda_{\text{frob}} \|\mathbf{C} - \mathbf{C}_s\|_F^2. \tag{9}$$

The second term is a regularization on how much the concept vectors can deviate from their source domain values in terms of the Frobenius norm.

## 3.2 LINEAR PROBING ADAPTATION

In this step, we focus on improving the test accuracy of the label predictor of the main CBM branch $(\mathbf{W}, \mathbf{b})$, with the concept vectors $\mathbf{C}$ fixed at their updated values from the CSA step (the residual CBM parameters are also frozen). For this, we use the cross-entropy loss between the predictions of the target domain CBM (Eqn. 5) and the pseudo-labels of a test batch $\widehat{\mathcal{D}}_t^b$. In order to enhance the interpretability of the label predictor, we impose sparsity and grouping effect in its weights via an Elastic-net penalty term (Zou & Hastie, 2005; Yuksekgonul et al., 2023) given by

$$L_{\text{sparse}}(\mathbf{W}) = \frac{1}{m L} \sum_{\ell=1}^{L} \left( \alpha \|\mathbf{w}_\ell\|_1 + (1 - \alpha) \|\mathbf{w}_\ell\|_2^2 \right), \tag{10}$$

where $\mathbf{w}_\ell \in \mathbb{R}^m$ is the $\ell$-th row of $\mathbf{W}$, and $\alpha = 0.99$. The adaptation objective for LPA is given by

$$L_{\text{LPA}}(\mathbf{W}, \mathbf{b}) = -\frac{1}{|\widehat{\mathcal{D}}_t^b|} \sum_{(\mathbf{x}_t, \widehat{y}_t) \in \widehat{\mathcal{D}}_t^b} \log \boldsymbol{\sigma}_{\widehat{y}_t}(\mathbf{f}_t^{(\text{cbm})}(\mathbf{x}_t)) + \lambda_{\text{sparse}} L_{\text{sparse}}(\mathbf{W}), \tag{11}$$

where $\boldsymbol{\sigma}_k(\mathbf{r})$ is the Softmax probability for class $k$ given the logits $\mathbf{r}$, and $\lambda_{\text{sparse}} \geq 0$ is a sparsity regularization hyper-parameter. Using this objective, the label predictor is adapted such that the CBM's predictions on a test batch are consistent with their pseudo-labels.

## 3.3 RESIDUAL CONCEPT BOTTLENECK

We next discuss adaptation of the residual branch of the CBM whose parameters are $\{\widetilde{\mathbf{C}}, \widetilde{\mathbf{W}}, \widetilde{\mathbf{b}}\}$. The $r$ additional concept vectors in $\widetilde{\mathbf{C}}$ are expected to capture new concepts in the target data and compensate for the potentially incomplete coverage of the main CBM (see Section 2.3). By increasing the expressiveness of the concept subspace, we expect to improve the accuracy on the target dataset beyond the CSA and LPA steps. Therefore, we first have a cross-entropy loss term in this adaptation objective (as in Eqn. 11). We also introduce a *cosine similarity* based regularization in the objective to encourage the new concept vectors in $\widetilde{\mathbf{C}}$ to be less redundant with each other, and to have less overlap with the existing concept vectors $\mathbf{C}$ (obtained from the CSA step).

$$L_{\text{sim}}(\widetilde{\mathbf{C}}) = \frac{1}{m r} \sum_{i \in [m]} \sum_{j \in [r]} \cos(\mathbf{c}_i, \widetilde{\mathbf{c}}_j) + \frac{2}{r (r-1)} \sum_{\substack{(i,j) \in [r]^2: \\ j > i}} \cos(\widetilde{\mathbf{c}}_i, \widetilde{\mathbf{c}}_j). \tag{12}$$

Finally, we include a *coherency regularization* term in the objective (modified from Yeh et al. (2020)) to improve the interpretability of the learned residual concepts, given by

$$L_{\text{coh}}(\widetilde{\mathbf{C}}) = \frac{1}{r k} \sum_{i \in [r]} \sum_{\mathbf{x}_t \in T_{\widetilde{\mathbf{c}}_i}} \frac{\langle \widetilde{\mathbf{c}}_i, \phi(\mathbf{x}_t) \rangle}{\|\widetilde{\mathbf{c}}_i\|_2}, \tag{13}$$

where $T_{\widetilde{\mathbf{c}}_i}$ is the subset of the current target batch $\mathcal{D}_t^b$ that has the $k$-largest concept scores for residual concept vector $\widetilde{\mathbf{c}}_i$ (*i.e.*, the top-$k$ nearest neighbors of $\widetilde{\mathbf{c}}_i$ among the feature representations from $\mathcal{D}_t^b$).

The objective to be minimized for adapting the residual concept bottleneck (with the parameters of the main CBM branch frozen) is given by:

$$L_{\text{RCB}}(\widetilde{\mathbf{C}}, \widetilde{\mathbf{W}}, \widetilde{\mathbf{b}}) = -\frac{1}{|\widehat{\mathcal{D}}_t^b|} \sum_{(\mathbf{x}_t, \widehat{y}_t) \in \widehat{\mathcal{D}}_t^b} \log \boldsymbol{\sigma}_{\widehat{y}_t}(\mathbf{f}_t^{(\text{cbm})}(\mathbf{x}_t)) + \lambda_{\text{sim}} L_{\text{sim}}(\widetilde{\mathbf{C}}) - \lambda_{\text{coh}} L_{\text{coh}}(\widetilde{\mathbf{C}}). \tag{14}$$

The constants $\lambda_{\text{sim}} \geq 0$ and $\lambda_{\text{coh}} \geq 0$ are hyper-parameters that control the strength of the regularization terms. Note that for the residual CBM, we *jointly* adapt $\widetilde{\mathbf{C}}$ and $\widetilde{\mathbf{W}}, \widetilde{\mathbf{b}}$, because we have a common objective of increasing the test accuracy, whereas for the main CBM, adaptation is done in two stages (CSA and LPA), with CSA focusing on distribution alignment of the concept scores based on the intra-class and inter-class distances. Additional details on our method are given in Appendix B. This includes 1) complexity analysis and evaluation of running times of CONDA; and 2) automatic annotation (captioning) of the adapted and residual concept vectors for interpretability analysis.

## 4 EXPERIMENTS

In this section, we conduct experiments to answer the following three research questions:

**RQ1:** How effective is CONDA in improving the test-time performance of deployed classification pipelines that use a foundation models with a concept bottleneck predictor?

**RQ2:** How does each component of CONDA specifically address and remedy the failures caused by different types of distribution shifts?

**RQ3:** How do the concept-based explanations change before and after test-time adaptation?

### 4.1 SETUP

**Datasets.** We evaluate the performance of concept bottlenecks for FMs and the proposed adaptation on five real-world datasets with distribution shifts, following the setup in Lee et al. (2023): (1) CIFAR10 to CIFAR10-C and CIFAR100 to CIFAR100-C for low-level shift, (2) Waterbirds and Metashift for concept-level shift, and (3) Camelyon17 for natural shift.

**Backbone Foundation Models.** For the CIFAR datasets, we use CLIP:ViT-L/14 (FARE[2]) (Schlarmann et al., 2024), which is adversarially fine-tuned to be more robust to (adversarial) low-level perturbations than standard CLIP variants. We employ CLIP:ViT-L/14 (Radford et al., 2021) for Waterbirds and Metashift. For Camelyon17, we utilize BioMedCLIP (Zhang et al., 2023), which is pre-trained on diverse medical domains to understand medical images and text jointly, making it suitable for zero-shot tasks in the medical domain.

**Preparing the Concept Bottleneck.** We evaluate CONDA using three popular approaches for constructing the concept bottleneck: (1) using a general-purpose concept bank where natural language concept descriptions and modern vision-language models (*e.g.*, Stable Diffusion (Rombach et al., 2022)) are leveraged to automatically generate concept examples for finding concept vectors (Yuksekgonul et al., 2023; Wu et al., 2023b); (2) unsupervised learned concepts where concept vectors are learned via optimization to maximize the concept-based prediction accuracy (Yeh et al., 2020); and (3) employing GPT-3 with appropriate filtering to discover a tailored set of concepts for the bottleneck (Oikarinen et al., 2023). More details can be found in Appendix A and Appendix C.2.

**Metrics.** We report the performance in terms of two metrics: averaged group accuracy (AVG) and worst-group accuracy (WG). AVG is the average (per-class) accuracy across the classes, and WG is the minimum (per-class) accuracy across the classes.

### 4.2 RQ1: EFFECTIVENESS OF CONDA UNDER REAL-WORLD DISTRIBUTION SHIFTS

Table 1 presents our main results evaluating the effectiveness of CONDA on different real-world distribution shifts, when combined with different CBM baselines. First of all, we observe that leveraging the expressive power of the FM feature representations can enhance the performance of CBMs. For example, using the method from Oikarinen et al. (2023), their reported accuracies on CIFAR10 and CIFAR100 are 86.40% and 65.13% respectively when using the CLIP-RN50 backbone. In our experiments, by employing the adversarially fine-tuned CLIP-ViT-L/14, we achieve higher accuracies of 95.24% and 68.36% respectively (source domain). This demonstrates the potential for improved utility in concept-based interpretable pipelines as foundation models continue to improve.

However, this improved performance in the source domain often does not translate to robustness after deployment. Under low-level shifts, the performance of CBMs may be comparable to that of

| Dataset | | | ZS | LP | Yuksekgonul et al. (2023) | | Yeh et al. (2020) | | Oikarinen et al. (2023) | |
|---|---|---|---|---|---|---|---|---|---|---|
| | | | | | Unadapted | w/ CONDA | Unadapted | w/ CONDA | Unadapted | w/ CONDA |
| CIFAR10 | Source | AVG | 91.18 | 93.26 ± 0.02 | 92.55 ± 0.05 | - | 96.26 ± 0.11 | - | 95.24 ± 0.08 | - |
| | | WG | 71.1 | 88.23 ± 0.08 | 85.64 ± 0.55 | - | 90.89 ± 0.97 | - | 90.11 ± 0.76 | - |
| | Target | AVG | 66.68 ± 15.88 | 84.11 ± 1.54 | 82.61 ± 1.65 | **84.38 ± 1.52** | 89.76 ± 1.10 | 85.14 ± 1.29 | 81.22 ± 2.77 | **84.56 ± 3.11** |
| | | WG | 55.04 ± 2.05 | 71.37 ± 3.33 | 68.62 ± 2.93 | **72.69 ± 2.49** | 78.28 ± 2.43 | 76.09 ± 1.66 | 69.03 ± 2.47 | **72.88 ± 2.01** |
| CIFAR100 | Source | AVG | 62.73 | 66.67 ± 0.29 | 65.98 ± 0.10 | - | 83.87 ± 0.04 | - | 68.36 ± 0.09 | - |
| | | WG | 5.12 | 4.28 ± 0.51 | 9.5 ± 1.14 | - | 51.0 ± 1.40 | - | 12.09 ± 1.23 | - |
| | Target | AVG | 51.90 ± 1.76 | 55.30 ± 1.63 | 51.53 ± 0.13 | **53.88 ± 0.23** | 72.33 ± 0.15 | 70.82 ± 0.20 | 52.16 ± 0.14 | **54.79 ± 1.17** |
| | | WG | 1.73 ± 0.4 | 2.47 ± 0.49 | 2.80 ± 0.71 | 2.56 ± 0.27 | 30.60 ± 1.42 | 28.44 ± 0.95 | 6.32 ± 0.38 | 6.01 ± 0.22 |
| Waterbirds | Source | AVG | 82.61 | 97.43 ± 0.05 | 97.78 ± 0.16 | - | 98.80 ± 0.04 | - | 98.80 ± 0.17 | - |
| | | WG | 67.45 | 95.08 ± 0.11 | 96.31 ± 0.38 | - | 98.21 ± 0.08 | - | 97.03 ± 0.26 | - |
| | Target | AVG | 61.06 | 54.10 ± 0.55 | 32.03 ± 0.58 | **60.69 ± 0.23** | 45.03 ± 0.34 | **61.11 ± 0.09** | 46.18 ± 0.42 | **62.71 ± 0.33** |
| | | WG | 42.52 | 44.70 ± 0.70 | 27.80 ± 1.24 | **43.01 ± 0.46** | 38.74 ± 0.68 | **41.86 ± 0.25** | 35.29 ± 1.52 | **44.01 ± 0.60** |
| Metashift | Source | AVG | 95.72 | 97.27 ± 0.28 | 97.94 ± 0.10 | - | 97.18 ± 0.01 | - | 98.02 ± 0.10 | - |
| | | WG | 93.44 | 96.62 ± 0.39 | 96.94 ± 0.30 | - | 96.0 ± 0.01 | - | 97.25 ± 0.10 | - |
| | Target | AVG | 94.65 | 80.39 ± 0.42 | 84.45 ± 1.39 | **93.69 ± 0.20** | 90.53 ± 0.09 | **93.81 ± 0.13** | 83.72 ± 2.21 | **93.90 ± 0.13** |
| | | WG | 92.81 | 65.33 ± 0.61 | 73.89 ± 3.21 | **92.02 ± 0.12** | 84.84 ± 0.20 | **91.41 ± 0.26** | 75.41 ± 1.68 | **91.77 ± 0.12** |
| Camelyon17 | Source | AVG | 77.71 | 92.14 ± 0.01 | 89.07 ± 0.60 | - | 97.01 ± 0.05 | - | 94.19 ± 0.11 | - |
| | | WG | 69.73 | 88.89 ± 0.02 | 84.34 ± 1.39 | - | 96.31 ± 0.24 | - | 91.23 ± 0.12 | - |
| | Target | AVG | 84.55 | 93.69 ± 0.01 | 89.71 ± 0.65 | **91.20 ± 0.06** | 95.01 ± 0.07 | 92.54 ± 0.16 | 91.75 ± 0.08 | **93.16 ± 0.05** |
| | | WG | 76.08 | 89.49 ± 0.02 | 85.96 ± 0.88 | **88.96 ± 0.16** | 93.07 ± 0.37 | 91.07 ± 0.32 | 87.24 ± 0.09 | **89.00 ± 0.07** |

Table 1: **Performance of CONDA on different distribution shifts when combined with different CBMs.** Zero-shot (ZS) and Linear probing (LP) are the non-interpretable FM baselines. Low-level shifts are covered by the CIFAR datasets, concept-level shifts by Waterbirds and Metashift, and natural shifts by the Camelyon17 benchmark. CONDA significantly improves the AVG and WG accuracy on the target domain in many scenarios.

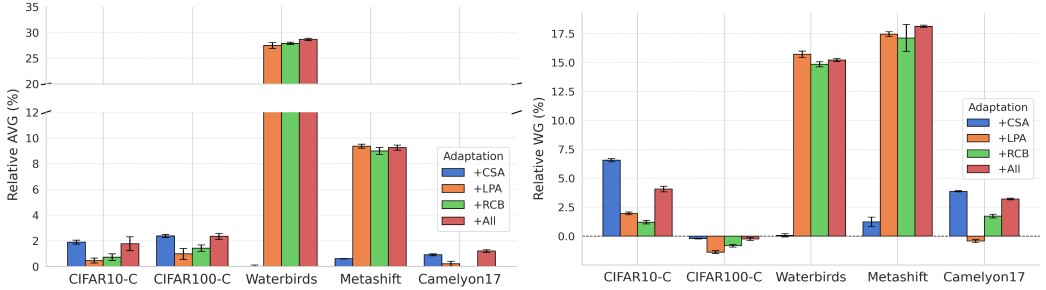

Figure 3: **Effectiveness of individual components of CONDA for the CBM method of Yuksekgonul et al. (2023)**. We report the relative AVG and WG, which is the (acc. after adaptation) − (acc. before adaptation).

the non-interpretable counterparts (ZS and LP), but they are not inherently more robust to low-level shifts. The performance drop is particularly severe under concept-level shifts when the CBM is not adapted. But with adaptation using CONDA, the test-time accuracy under different distribution shifts increases significantly in most cases. The performance is on par with or even surpasses that of the non-interpretable methods, notably in terms of the WG accuracy.

### 4.3 RQ2: Effectiveness of Individual Components of CONDA

We next analyze the individual contributions of the components in CONDA, *viz.* CSA, LPA, and RCB. Figure 3 illustrates the relative AVG and WG (%) when adapting the CBM of Yeh et al. (2020). Under low-level shifts, CSA plays a crucial role in performance improvement by encouraging the high-level concept scores to remain similar. Interestingly, using CSA alone even surpasses the performance achieved when all components are combined. This trend is also observed with the Camelyon17 dataset, which resembles a low-level shift due to lighting differences across hospitals. On the other hand, under concept-level shifts, LPA and RCB become the key components of adaptation. These components allow the model to adjust concept reliance to the target domain and address the incompleteness of the deployed concept set, tailoring it to the target data. In this context, CSA has minimal impact, while using *only LPA* leads to performance gains comparable to, or even exceeding that achieved when all components are included.

This phenomenon aligns with the findings of Lee et al. (2023) that fine-tuning only a subset of layers can be more effective than fine-tuning all layers, depending on the type of distribution shift. In our case, the concept-based prediction pipeline can be considered a special instance of their framework with a two-layer classifier. The concept bottleneck layer corresponds to the first layer, which is

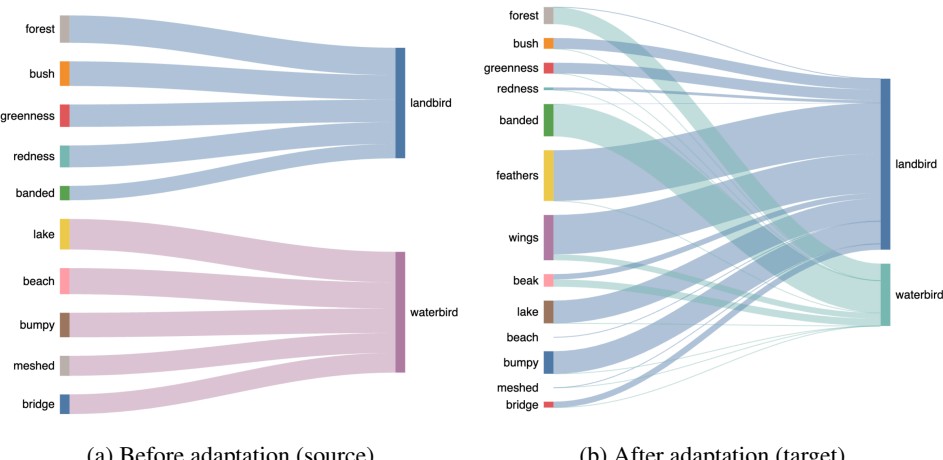

|                          |                          |
|:------------------------:|:------------------------:|
| (a) Before adaptation (source) | (b) After adaptation (target) |

Figure 4: **CONDA adapts the concept weights to be tailored to the target data.** We visualize the positive weights in the linear probing layer (width of each mapping) before vs. after applying CONDA to the PCBM baseline (Yuksekgonul et al., 2023) on the Waterbirds dataset.

particularly important for addressing input-level shifts (following their definition), while the linear probing layer corresponds to the second layer, which is more important for handling output-level shifts (see Section 3 of their paper). These empirical observations confirm our design motivation for CONDA, *i.e.*, different components play key roles in adapting to different types of distribution shifts.

Additional results, including ablation experiments to understand the effect of hyper-parameters, improved pseudo-labeling methods, and the choice of backbone foundation model are given in Appendix D. They provide additional support for the strong empirical performance of CONDA.

## 4.4 RQ3: INTERPRETABILITY OF CONDA

We investigate how the concept-based explanations change through adaptation by CONDA on the Waterbirds dataset. In Figure 4a, we present the top five most prominent concepts contributing to the predictions for each class. As expected, in the source domain, land-related concepts are most important for predicting "landbird", and do not positively contribute to "waterbird"; and vice versa for water-related concepts. After adapting to the target domain (test dataset), we observe adjustments in the *concept-to-class mappings*. Notably, land-related concepts begin to positively contribute to the prediction of "waterbird". This shift indicates that CONDA successfully adapts the concept-based explanations to reflect the new correlations observed in the target domain. Moreover, in the original concept bottleneck constructed following Wu et al. (2023b), there were no bird-related concepts that could help make robust predictions independent of spurious background correlations. By employing RCB with five residual concepts, we identified that three of them correspond to bird-related concepts: feathers, wings, and beak [5]. This demonstrates that CONDA adapts in a manner aligned with human intuition, just like a human intervening in a CBM to correct its predictions would. More importantly, RCB captures concepts that may have been missed during the initial construction of the concept bottleneck, enhancing both the interpretability and robustness. Additional results and analysis of the interpretability of CONDA can be found in Appendix E.

## 5 CONCLUSIONS AND FUTURE WORK

In this work, we made a first effort at exploring the test-time (post-deployment) performance of CBMs combined with foundation models. We formalized potential failure modes under low-level and concept-level distribution shifts and proposed a novel test-time adaptation framework. Each component of our framework is designed to address specific failure modes, effectively improving the test-time performance of a deployed CBM. Limitations and future work are discussed in Appendix F.

---

[5]To interpret the residual concepts, we use automated concept annotations; see details in Appendix B.2.

## ACKNOWLEDGMENTS

Jihye Choi, Jayaram Raghuram and Somesh Jha are partially supported by DARPA under agreement number 885000, NSF CCF-FMiTF-1836978 and ONR N00014-21-1-2492. Yixuan Li is supported by the AFOSR Young Investigator Program under award number FA9550-23-1-0184, NSF IIS-2237037, NSF IIS-2331669, and Office of Naval Research under grant number N00014-23-1-2643.

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

# APPENDICES

| Symbol | Description |
|---|---|
| $d$ | Dimension of the feature representation from the foundation model |
| $L$ | Number of classes or size of the label set |
| $m$ | Number of concepts in the main CBM |
| $r$ | Number of concepts in the residual CBM |
| $\mathbf{x}_s, y_s$ | Input and corresponding label in the source domain |
| $\mathbf{x}_t, \widehat{y}_t$ | Input and corresponding pseudo-label in the target domain |
| $\phi(\mathbf{x})$ | Foundation model or the backbone feature extractor |
| $D_s$ and $D_t$ | Source and target domain |
| $\mathbf{h} : \mathbb{R}^d \to \mathbb{R}^m$ with $\mathbf{h} \in \mathcal{H}$ | Concept mapping and concept hypothesis class |
| $\mathbf{g} : \mathbb{R}^m \to \mathbb{R}^L$ with $\mathbf{g} \in \mathcal{G}$ | Classifier and classification hypothesis class |
| $\mathbb{P}_{\text{con}}(D_j, \phi, \mathbf{h})$ | Concept score distribution for domain $D_j$, $j \in \{s, t\}$ |
| $\mathbb{P}_{\text{pred}}(D_j, \phi, \mathbf{h}, \mathbf{g})$ | Prediction distribution for domain $D_j$, $j \in \{s, t\}$ |
| $\mathbf{C}$ | Matrix of concept vectors adapted for the target domain of size $m \times d$. Subscripts 's' and 't' refer to the source and target domain respectively. |
| $\widetilde{\mathbf{C}}$ | Matrix of residual concept vectors adapted for the target domain of size $r \times d$ |
| $\mathbf{W}, \mathbf{b}$ | Main CBM linear predictor parameters. $\mathbf{W}$ has size $L \times m$ and $\mathbf{b}$ has length $L$ |
| $\widetilde{\mathbf{W}}, \widetilde{\mathbf{b}}$ | Residual CBM linear predictor parameters. $\widetilde{\mathbf{W}}$ has size $L \times r$ and $\widetilde{\mathbf{b}}$ has length $L$ |
| $\mathbf{f}_s^{(\text{cbm})}(\mathbf{x})$ | CBM predictor in the source domain. See Eqn. (1) |
| $\mathbf{f}_t^{(\text{cbm})}(\mathbf{x})$ | CBM predictor in the target domain. See Eqn. (5) |
| $\mathcal{D}_t$ | Unlabeled test dataset |
| $\mathcal{D}_t^b$ and $\widehat{\mathcal{D}}_t^b$ | Test data batch. First one is unlabeled, while the second one includes the pseudo-labels. |
| $\boldsymbol{\mu}_y, \boldsymbol{\Sigma}_y$ | Class-specific mean and covariance matrix of the concept scores from the source dataset |
| $\mathbf{v}_{\mathbf{C}}(\mathbf{x})$ | Concept scores obtained from the concept vectors in $\mathbf{C}$ via the projection $\mathbf{C}\phi(\mathbf{x})$ |
| $D_{\text{mah}}(\mathbf{x}; \boldsymbol{\mu}_y, \boldsymbol{\Sigma}_y)$ | Mahalanobis distance in the concept-score space |
| $\boldsymbol{\sigma}_k(\mathbf{r})$ | Softmax probability for class $k$ given the logits $\mathbf{r}$ |
| $\ell_{ada}(\cdot)$ | Adaptation loss used for feature alignment. See Eqn. (8) |
| $L_{\text{CSA}}$ | Adaptation objective for CSA. See Eqn. (9) |
| $L_{\text{sparse}}$ | Elastic-net penalty regularization used in LPA. See Eqn. (10) |
| $L_{\text{LPA}}$ | Adaptation objective for LPA. See Eqn. (11) |
| $L_{\text{sim}}$ | Cosine similarity based regularization used in RCB. See Eqn. (12) |
| $L_{\text{coh}}$ | Coherancy regularization used in RCB. See Eqn. (13) |
| $L_{\text{RCB}}$ | Adaptation objective for RCB. See Eqn. (14) |
| $n_{\text{grad}}$ | Number of gradient steps for each of the CSA, LPA, and RCB adaptations. |
| $n_{\text{batch}} = |\mathcal{D}_t^b|$ | Batch size of adaptation |

Table 2: Symbols and notations used in the paper.

---

**Algorithm 1** CONDA: CONCEPT-BASED DYNAMIC ADAPTATION

---

**Inputs:** Foundation model $\phi(\mathbf{x})$. Source domain CBM: $\mathbf{C}_s, \mathbf{W}_s, \mathbf{b}_s$. Concept scores distribution statistics: $\{(\boldsymbol{\mu}_y, \boldsymbol{\Sigma}_y)\}_{y \in \mathcal{Y}}$. Unlabeled test dataset $\mathcal{D}_t$.

1: **Set constants and hyper-parameters:**

    # batches $B$, # gradient steps $n_{\text{grad}}$, # residual concepts $r$
    Regularization constants: $\lambda_{\text{frob}}, \lambda_{\text{sparse}}, \lambda_{\text{sim}}, \lambda_{\text{coh}}$

2: Initialize the main CBM branch using source domain parameters: $\mathbf{C} = \mathbf{C}_s, \mathbf{W} = \mathbf{W}_s, \mathbf{b} = \mathbf{b}_s$.
3: Initialize the residual CBM branch parameters $\widetilde{\mathbf{C}}, \widetilde{\mathbf{W}}, \widetilde{\mathbf{b}}$ randomly.
4: Split the test dataset randomly into $B$ fixed-size batches $\{\mathcal{D}_t^b\}_{b=1}^B$.

5: **for** batch $b = 1, 2, \cdots, B$ **do**

6:    **Pseudo-labeling:** Using the foundation model, take an ensemble of the zero-shot predictor and the linear-probing predictor to obtain pseudo-labels for the test batch. More advanced methods can be used here, *e.g.*, the soft nearest-neighbor voting of Chen et al. (2022).

7:    **CSA Step:** Adapt $\mathbf{C}$ with the remaining parameters fixed at their current values.
8:    **for** step $i = 1, 2, \cdots, n_{\text{grad}}$ **do**
9:        Compute the intra-class and inter-class Mahalanobis distances for the pseudo-labeled test batch $\widehat{\mathcal{D}}_t^b$ (Eqns. 6 and 7).
10:       Compute the CSA adaptation objective $L_{\text{CSA}}(\mathbf{C})$ (Eqns. 8 and 9).
11:       Perform a gradient descent step to update $\mathbf{C}$.
12:    **end for**

13:    **LPA Step:** Adapt $(\mathbf{W}, \mathbf{b})$ with the remaining parameters fixed at their current values.
14:    **for** step $i = 1, 2, \cdots, n_{\text{grad}}$ **do**
15:       Compute the Elastic-net regularization term $L_{\text{sparse}}(\mathbf{W})$ (Eqn. 10).
16:       Compute the LPA adaptation objective $L_{\text{LPA}}(\mathbf{W}, \mathbf{b})$ (Eqn. 11).
17:       Perform a gradient descent step to update $\mathbf{W}, \mathbf{b}$.
18:    **end for**

19:    **RCB Step:** Adapt $(\widetilde{\mathbf{C}}, \widetilde{\mathbf{W}}, \widetilde{\mathbf{b}})$ with the remaining parameters fixed at their current values.
20:    **for** step $i = 1, 2, \cdots, n_{\text{grad}}$ **do**
21:       Compute the cosine similarity regularization term $L_{\text{sim}}(\widetilde{\mathbf{C}})$ (Eqn. 12).
22:       Compute the coherency regularization term $L_{\text{coh}}(\widetilde{\mathbf{C}})$ (Eqn. 13).
23:       Compute the RCB adaptation objective $L_{\text{RCB}}(\widetilde{\mathbf{C}}, \widetilde{\mathbf{W}}, \widetilde{\mathbf{b}})$ (Eqn. 14).
24:       Perform a gradient descent step to update $\widetilde{\mathbf{C}}, \widetilde{\mathbf{W}}, \widetilde{\mathbf{b}}$.
25:    **end for**

26:    Using the adapted parameters, obtain the target domain CBM predictions $\mathbf{f}_t^{(\text{cbm})}(\mathbf{x})$ for the current batch (Eqn. 5).
27:    Initialize parameters for the next batch using the adapted parameters from the current batch.

28: **end for**

---

**Outputs:** Predictions of the target domain CBM on the test dataset. Final adapted parameters of the target domain CBM: $\mathbf{C}_t, \mathbf{W}_t, \mathbf{b}_t, \widetilde{\mathbf{C}}_t, \widetilde{\mathbf{W}}_t, \widetilde{\mathbf{b}}_t$.

---

## A    EXPANDED RELATED WORK

Concept Bottleneck Models (CBMs), introduced by Koh et al. (2020), are interpretable neural networks that map input data to a set of human-understandable concepts (the "bottleneck") before making predictions. This architecture enhances the interpretability by revealing which concepts influence the predictions and allows users to intervene by adjusting mis-predicted concepts.

**Definition of Concept Bottleneck.** Despite the above benefits, early variants (*e.g.*, (Havasi et al., 2022)) required extensive concept annotations during training, which can be costly and impractical. This reliance on predefined, annotated concepts limits their scalability and applicability to diverse domains and tasks. To address this, recent methods aim to construct CBMs without requiring explicit concept labels, and they can be placed into three main categories: (a) unsupervised learning-based concept discovery, (b) general-purpose concept bank agnostic to tasks, and (c) leveraging multi-modal foundation models. They are further discussed below.

(a)  Unsupervised learning-based concept discovery: Yeh et al. (2020) formulates the concept discovery as an optimization process with the objective of concept completeness, ensuring that the extracted concepts comprehensively represent the data while maintaining interpretability. This approach is further advanced in Wang et al. (2023), where they optimize task-specific concepts via self-supervision techniques such as contrastive loss to improve the quality of the learned concepts.

(b)  General-purpose concept bank agnostic to tasks: Yuksekgonul et al. (2023) and Wu et al. (2023b) utilize a predefined concept bank where each concept vector is derived from the parameters of a Support Vector Machine (SVM) trained to distinguish between positive and negative instances in image embeddings obtained from a backbone model. Here the dataset used to learn the SVMs does not have to be the same as the data for the given task.

(c)  Leveraging multi-modal foundation models: Another approach leverages the rapid advancements in multi-modal foundation models like CLIP to align visual and textual representations, enabling the mapping of each concept to a human-readable description (Moayeri et al., 2023). Yuksekgonul et al. (2023) also suggests defining each concept vector with the text embeddings from the backbone, where the text serves as human-understandable concept descriptions (refer to Figure 2 in Shang et al. (2024) for a descriptive illustration of the method). Oikarinen et al. (2023) relies on a pre-trained backbone like CLIP which maps images and textual descriptions into a shared embedding space. They define each concept vector as the mapping of an image embedding to its corresponding text embedding.

In our paper, we consider the most representative method from each category of concept bottleneck constructions: Yeh et al. (2020) for (a), Yuksekgonul et al. (2023) for (b), and Oikarinen et al. (2023) for (c) (refer to Appendix C.2 for further details on their implementations). By applying our adaptation framework to various definitions of a concept bottleneck, we demonstrate that it can effectively and flexibly enhance the post-deployment robustness of various CBM types under real-world distribution shift scenarios.

**Concept-based explanations and distribution shifts.** There has been growing interest in the utility of concept-based explanations under distribution shifts. The initial work by (Kim et al., 2018) hinted at the potential of high-level concepts as diagnostic units against low-level perturbations, such as adversarial examples. Following this, Adebayo et al. (2020) suggested that concept-based explanations could be more robust tools for debugging and analyzing model behaviors under spurious correlations. More recently, Abid et al. (2022) and Wu et al. (2023b) have studied the utility of concept-based explanations in the context of data drift. However, these works rely on a predefined concept bank that remains *static* after model deployment. Our work emphasizes the need for a *dynamic* approach to concept bottlenecks for the optimal utility of concept-based predictions in the deployment phase where test data can have distribution shifts. To the best of our knowledge, this is the first work to present a comprehensive view of the post-deployment performance of concept-based prediction pipelines, and to address their test-time adaptation under distribution shifts with a dynamic concept bank.

# B  ADDITIONAL METHOD DETAILS

We describe the comprehensive algorithm of CONDA in Algorithm 1.

## B.1  COMPLEXITY ANALYSIS OF CONDA

Referring to Algorithm 1, we evaluate the computational complexity of the CSA, LPA, RCB, and pseudo-labeling steps for a single test batch $\mathcal{D}_t^b$. We recall some of the terms used in our notation.

- $m$ : number of concepts in the main CBM.
- $r$ : number of concepts in the residual CBM branch.
- $d$ : dimension of the feature representation $\phi(\mathbf{x})$.
- $L$ : number of classes.
- $n_{\text{grad}}$ : number of gradient steps for each of the CSA, LPA, and RCB adaptations.
- $n_{\text{batch}} = |\mathcal{D}_t^b|$ : batch size

**CSA step** optimizes the concept matrix of the main CBM branch $\mathbf{C}$, which has $m\,d$ parameters. Below we breakdown the computations involved in the CSA adaptation objective and its gradient updates. We assume that the mean and inverse-covariance matrices of the class-conditional concept score distributions are pre-computed from the source dataset.

Concept score projection for a single test sample: $2\,m\,d$.

Intra-class and inter-class Mahalanobis distances for a single test sample: $L\,(2m^2 + 3m) + L = L\,m\,(2\,m + 3) + L$.

Frobenius norm regularization term: $3\,m\,d$.

Cost of stochastic gradient update step for the batch: $2\,m\,d$.

The cost of optimizing the CSA objective (Eqn. 9) for $n_{\text{grad}}$ gradient update steps can be expressed as:

$$\text{Cost}_{\text{CSA}} = n_{\text{grad}}\Big(n_{\text{batch}}\,\big(2\,m\,d + L\,m\,(2\,m + 3) + L\big) + 5\,m\,d\Big). \tag{15}$$

**LPA step** optimizes the linear predictor in the main CBM branch, whose parameters are $(\mathbf{W}, \mathbf{b})$. The number of parameters optimized in this step is $L\,(m + 1)$. Below we breakdown the computations involved in the LPA adaptation objective and its gradient updates.

Elastic-Net regularization term: $3\,L\,m$.

Cross-entropy loss term in the LPA adaptation objective (Eqn. 11) for a single test sample: $2\,m\,d + 2\,L\,m + L + 2\,r\,d + 2\,L\,r + L + 3\,L = 2\,(m + r)\,(d + L) + 5\,L$.

Cost of stochastic gradient update step for the batch: $2\,L\,(m + 1)$.

The cost of optimizing the LPA objective for $n_{\text{grad}}$ gradient update steps can be expressed as:

$$\text{Cost}_{\text{LPA}} = n_{\text{grad}}\Big(n_{\text{batch}}\,\big(2\,(m + r)\,(d + L) + 5\,L\big) + 5\,L\,m + 2\,L\Big). \tag{16}$$

**RCB step** optimizes the concept matrix and linear predictor in the residual CBM branch, whose parameters are $(\widetilde{\mathbf{C}}, \widetilde{\mathbf{W}}, \widetilde{\mathbf{b}})$. The number of parameters optimized in this step is $r\,d + L\,(r + 1)$. Below we breakdown the computations involved in the RCB adaptation objective and its gradient updates.

Cosine similarity regularization term: $6\,d\,r\,(m + (r - 1)/2)$.

Coherancy regularization term: $r\,\big(4\,n_{\text{batch}}\,d + n_{\text{batch}}\,\log(n_{\text{batch}}) + k\big)$.

Cross-entropy loss term in the RCB adaptation objective (Eqn. 14) for a single test sample: $2\,m\,d + 2\,L\,m + L + 2\,r\,d + 2\,L\,r + L + 3\,L = 2\,(m + r)\,(d + L) + 5\,L$.

Cost of stochastic gradient update step for the batch: $2\,r\,d + 2\,L\,(r + 1)$.

The cost of optimizing the RCB objective for $n_{\text{grad}}$ gradient update steps can be expressed as:

$$\text{Cost}_{\text{RCB}} = n_{\text{grad}} \Big( n_{\text{batch}} \left( 2 \left( m + r \right) \left( d + L \right) + 5 L \right) + 6 \, d \, r \left( m + (r-1)/2 \right)$$

$$+ 4 \, n_{\text{batch}} \, r \, d + r \, n_{\text{batch}} \log(n_{\text{batch}}) + k \, r + 2 \, r \, d + 2 L \left( r + 1 \right) \Big) \quad (17)$$

**Pseudo-labeling.** For the pseudo-labeling method based on the ensemble of the zero-shot predictor and linear-probing predictor with the foundation model backbone, the computational complexity will be dominated by the architecture and number of parameters $p$ in the foundation model. We denote the inference cost on a test batch by $C_\phi(p, n_{\text{batch}})$. The computation involved in the zero-shot and linear probing predictions will be negligible compared to this.

The overall computational cost for a single test batch is the sum of the costs of the CSA, LPA, RCB, and pseudo-labeling steps described above. From this, we select the terms that dominate the computational cost and ignore terms that depend on smaller quantities like $r$ and $k$. This leads us an **overall complexity** of $\mathcal{O}\big(n_{\text{grad}} \, n_{\text{batch}} \, m^2 \, L + n_{\text{grad}} \, n_{\text{batch}} \, m \, d + n_{\text{grad}} \, d \, r \left( m + r \right)\big)$. The number of concepts $m$ is usually on the order of hundreds and $r$ is much smaller than that. The embedding dimension $d$ is on the order of few hundreds to thousands, depending on the foundation model.

To quantify the computational complexity of our method, in Table 3, we report the average (per-batch) wall-clock running times of CONDA when combined with post-hoc CBM (Yuksekgonul et al., 2023). We observe that adaptation using CONDA is quite fast, taking only a few seconds per batch, with the main time-consuming component being the pseudo-labeling since it involves inference on the foundation model.

| Backbone | Embedding size | Target Dataset | Dataset size | PL | +CSA | +LPA | +RCB | All |
|---|---|---|---|---|---|---|---|---|
| CLIP:ViT-L-14 (FARE[2]) | 768 | CIFAR10-C | 10000 | 4.113 | 0.116 | 0.021 | 0.038 | 4.288 |
| CLIP:ViT-L-14 (FARE[2]) | 768 | CIFAR100-C | 10000 | 16.203 | 0.692 | 0.023 | 0.041 | 16.959 |
| CLIP:ViT-L-14 | 768 | Waterbirds | 2897 | 0.064 | 0.043 | 0.028 | 0.051 | 0.186 |
| CLIP:ViT-L-14 | 768 | Metashift | 541 | 0.077 | 0.051 | 0.022 | 0.039 | 0.188 |
| BiomedCLIP | 512 | Camelyon17 | 85054 | 0.387 | 0.089 | 0.042 | 0.078 | 0.596 |

Table 3: **Runtime of CONDA.** All inputs are reshaped to the dimension of (224, 224, 3). In column PL, we report the time it takes to obtain our proposed pseudo-labeling *per batch* (averaged across all incoming batches). We run 20 adaptation steps for each of CSA, LPA, and RCB with five residual concepts, and report runtime (in seconds) averaged across all batches at the test time.

### B.2 AUTOMATICALLY ANNOTATING CONCEPTS

We adopt and modify CLIP-DISSECT (Oikarinen & Weng, 2023) for automatically annotating the concepts as follows.

Suppose $\mathcal{S}$ is the set of possible concept annotations. We use `ConceptNet` (Speer et al., 2017) to obtain texts that are relevant to the classes. ConceptNet is an open knowledge graph, where we can find concepts that have particular relationships to a query text. For instance, for a class "cat", one can find relations of the form "A Cat has {whiskers, four legs, sharp claws, ...}". Similarly, we can find "parts" of a given class (*e.g.*, "bumper", "roof" for "truck" class), or the superclass of a given class (*e.g.*, "animal", "canine" for "dog"). Following the setup in Yuksekgonul et al. (2023), we restrict ourselves to five sets of relations for each class: the `hasA`, `isA`, `partOf`, `HasProperty`, and `MadeOf` relations in ConceptNet. We collect all the concepts that have these relations with the classes in each classification task to build the concept annotation set. However, for the Waterbirds dataset, since the classes of {"waterbird", "landbird"} are too specific in their terminology and we cannot find relevant nodes in ConceptNet, we instead use {"bird", "water", "land"} as the query set. When we have the concept annotations for the main concept bottleneck from before-deployment (*e.g.*, (Yuksekgonul et al., 2023; Oikarinen et al., 2023; Wu et al., 2023b)), we set $\mathcal{S}$ as the union set of those pre-defined concepts and those identified by ConceptNet.

Let $\mathcal{D}_t$ be the target domain (test) dataset. Let $\phi_{\text{CLIP}}^I$ and $\phi_{\text{CLIP}}^T$ be the image encoder and text encoder (respectively) of CLIP:ViT-B/16. Recall that $\phi$ is the backbone foundation model used in

our framework. To determine the annotation for a concept vector $\mathbf{c}_a \in \mathbf{C}_t$, our goal is to assign to it the most relevant caption $t_b \in \mathcal{S}$ as follows:

1. Compute the normalized text embedding of the concepts in $\mathcal{S}$ using $\phi_{\text{CLIP}}^T$; let $\mathbf{T}_j$ be the normalized text embedding of the $j$-th concept in $\mathcal{S}$. Also, compute the image embedding of all images in $\mathcal{D}_t$ using $\phi_{\text{CLIP}}^I$; let $\mathbf{I}_i$ be the image embedding of the $i$-th image in $\mathcal{D}_t$. We then compute the inner product of the all pairs of image-text embeddings via the image-text matrix $\mathbf{P} = \mathbf{I}\,\mathbf{T}^\top \in \mathbb{R}^{|\mathcal{D}_t| \times |\mathcal{S}|}$ where $\mathbf{I} \in \mathbb{R}^{|\mathcal{D}_t| \times d}$ and $\mathbf{T} \in \mathbb{R}^{|\mathcal{S}| \times d}$ and $d$ is the dimension of the CLIP embeddings. That is, $P_{ij}$ is the inner product of the normalized embeddings of the $i$-th target image and the $j$-th candidate annotation.

2. For all images in the target dataset, we compute and collect their concept scores as $\mathbf{v}_{\mathbf{c}_a} = [\langle \phi(\mathbf{x}_1), \mathbf{c}_a \rangle, \cdots, \langle \phi(\mathbf{x}_{|\mathcal{D}_t|}), \mathbf{c}_a \rangle]^\top \in \mathbb{R}^{|\mathcal{D}_t|}$.

3. The annotation for $\mathbf{c}_a$ is determined by calculating the most *similar* concept label in $\mathcal{S}$ based on its concept scores $\mathbf{v}_{\mathbf{c}_a}$. The similarity with respect to a concept $t_\ell \in \mathcal{S}$ is defined as

$$\texttt{sim}(t_\ell, \mathbf{v}_{\mathbf{c}_a}; \mathbf{P}) = \frac{\langle \mathbf{v}_{\mathbf{c}_a}, \mathbf{P}_{:,\ell} \rangle}{\|\mathbf{v}_{\mathbf{c}_a}\| \, \|\mathbf{P}_{:,\ell}\|}, \tag{18}$$

which is the cosine similarity between the concept scores and the corresponding column $\ell$ of the image-text matrix $\mathbf{P}_{:,\ell}$. Then, the annotation for $\mathbf{c}_a$ becomes the concept in $\mathcal{S}$ with the maximum similarity, given by $t_b$ where $b = \arg\max_\ell \texttt{sim}(t_\ell, \mathbf{v}_{\mathbf{c}_a}; \mathbf{P})$. To reduce noise in the annotations, we only accept $t_b$ as the annotation for $\mathbf{c}_a$ only when $\texttt{sim}(t_b, \mathbf{v}_{\mathbf{c}_a}; \mathbf{P}) > 0.8$.

To annotate the concepts in the residual concept bottleneck $\widetilde{\mathbf{C}}$, we repeat the same process.

## C  EXPERIMENTAL DETAILS

All the experiments are run on a server with thirty-two AMD EPYC 7313P 883 16-core processors, 528 GB of memory, and four 884 Nvidia A100 GPUs. Each GPU has 80 GB of 885 memory. For each setup, we repeated each experiment for 10 trials (using seed 40–49 for the random number generation) and report the mean and standard error.

### C.1  DATASETS

**CIFAR10.** It consists of 60k RGB images of size 32x32 (50k images for the train set, and 10k images for the test set), equally balanced over 10 different classes (*e.g.*, airplane, car, dog, cat, etc.). We follow the given train/test split to report the performance in the source domain.

**CIFAR100.** It is similar to CIFAR10, but in a larger-scale; there are 100 classes, and each class has 500 32x32 RGB training images and 100 test images, making the classification more challenging.

**CIFAR10-C and CIFAR100-C.** To report the accuracies, we take the average over 15 different types of corruptions with the severity level of two (out of the scale from one to five); Gaussian Noise, Shot Noise, Impulse Noise, Defocus Blur, Frosted Glass Blur, Motion Blur, Zoom Blur, Snow, Frost, Fog, Brightness, Contrast, Elastic, Pixelate, JPEG Compression. Conventionally, studies in out-of-distribution generalization literature, severity level five is used, but we observe that it severely hurts the performance of the foundation model, making it impossible to be used as a decent oracle for the pseudo labeling. Hence, we chose the severity level two that still causes the performance drop due to the distribution shift, but against which, the backbone model still presents decent performance compared to the CBMs.

**Waterbirds.** Waterbirds dataset is for a two-class classification task ("landbird" vs. "waterbird"). In the source domain, landbird (waterbird) images are always associated with the land (water) background, while in the target domain, the correlation with the background is flipped, *i.e.*, landbird (waterbird) images are always on the water (land) background.

**Metashift.** Metashift has two classes of "cat" and "dog", and it simulates the disparate correlation to the backgrounds in a similar way. Source cat images are always correlated with a sofa or bed in the background, while dog images are always correlated with a bench or bike in the background. For

evaluation, we randomly split 90:10 equally across the correlation types, *i.e.*, 10% of dog images with sofa, 10% of dog images with bed, 10% of cat images with bench, and 10% of cat images with bike. In the target domain, both cat and dog images are always on the shelf background.

**Camelyon17.** This dataset is a collection of histopathology whole-slide images used for the detection of metastases in lymph nodes; classifying the given slide into benign tissue vs cancerous tissue. It includes images from five medical centers, each with different staining protocols, equipment, and imaging settings. These differences simulate natural real-world distribution shifts. We use the train set (hospital 1-3) for source, and the test set (hospital 5) for the target.

For zero-shot prediction, we use a basic text template: `"A photo of {class_name}"` for CIFAR10, CIFAR100, Waterbirds, and Metashift datasets. For the Camelyon17 dataset, we use the ensemble of prompts: for class benign, {`"A histopathology image of normal lymph node tissue stained with hematoxylin and eosin."`, `"An H&E stained slide showing healthy lymph node without cancer cells."`, `"Microscopic image of non-cancerous lymph node tissue."`, `"A pathology image of benign lymph node with normal histology."`, `"Hematoxylin and eosin stained section of normal lymph node.''`}; for class malicious, {`"A histopathology image of lymph node with metastatic breast cancer stained with hematoxylin and eosin."`, `"An H&E stained slide showing lymph node tissue infiltrated by cancer cells."`, `"Microscopic image of lymph node containing metastatic carcinoma."`, `"A pathology image of malignant lymph node with cancer metastasis."`, `"Hematoxylin and eosin stained section of lymph node with breast cancer metastases."`}.

### C.2 Preparing The Concept Bottleneck

There are various ways of defining the concept vectors $\{\mathbf{c}_{si}\}_{i=1}^{m}$ in the concept prediction layer $\mathbf{v}_{\mathbf{C}_s}(\mathbf{x})$. Early works on CBM required the training dataset to have concept annotations from domain experts in addition to the class labels for training the concept predictor (Koh et al., 2020). Subsequent works have also explored learning the concept vectors in an unsupervised manner (without any concept annotations) (Yeh et al., 2020; Choi et al., 2023). More recently, natural language concept descriptions and modern vision-language models (*e.g.*, Stable Diffusion (Rombach et al., 2022)) are being leveraged to automatically generate concept examples (Yuksekgonul et al., 2023; Wu et al., 2023b) for finding the Concept Activation Vectors (CAVs) (Kim et al., 2018) (each CAV corresponds to a $\mathbf{c}_{si}$), or to directly guide the construction of concept bank $\mathbf{C}_s$ (Oikarinen et al., 2023). We highlight that in all prior works (to our knowledge) the *concept bank remains static*, *i.e.*, once the set of concept vectors is defined and the CBM is deployed, its predictions are made based on these predefined concepts, regardless of any distribution shift at test time.

**Yuksekgonul et al. (2023).** For CIFAR10 and CIFAR100, we use the BRODEN visual concepts datasets Bau et al. (2017) to learn concept activation vectors, which are used to initialize the weights and bias parameters of the concept bottleneck layer, as described in Yuksekgonul et al. (2023). For Waterbirds and Metashift, we use the images belonging to the concept categories as follows; nature, color, and textures for Waterbirds, and nature, color, texture, city, household, and others for Metashift. For Camelyon17, we use color and textures categories, following the setting in Wu et al. (2023b).

**Yeh et al. (2020).** For a fair comparison, we set the number of the concepts to be the same as the size of concept bottleneck by Yuksekgonul et al. (2023) except with Metashift where we use 100 concepts instead, since with over 100 concepts, we found there are much unnecessary redundancy between them.

**Oikarinen et al. (2023).** Following their instructions, we create the initial concept set using GPT-3, followed by concept filtering. For the sparsity of the linear probing layer, we set $\lambda = 0.001$ and $\alpha = 0.5$.

Table 4 shows a summary of the major hyper-parameters used in our experiments. As for the hyper-parameter $k$ in Equation 13, we set $k$ equal to batch size $/ (2 \times$ number of classes), which is a heuristic that works well in practice.

| Dataset | Backbone | Batch Size | # Epochs | lr (CSA, LPA, RCB) | Adaptation steps | $\{\lambda_{\text{frob}}, \lambda_{\text{sparse}}, \lambda_{\text{sim}}, \lambda_{\text{coh}}\}$ |
|---------|----------|------------|----------|--------------------|------------------|------------------------------------------------|
| CIFAR10 | CLIP:ViT-L-14 (FARE$^2$) | 128 | 50 | Adam, 0.01 | 20 | $\{0.1, 1.0, 0.1, 2.0\}$ |
| CIFAR100 | CLIP:ViT-L-14 (FARE$^2$) | 512 | 50 | Adam, 0.01 | 20 | $\{0.1, 1.0, 0.1, 2.0\}$ |
| Waterbirds | CLIP:ViT-L-14 | 32 | 20 | SGD, 0.1 | 20 | $\{2.5, 1.0, 0.1, 0.1\}$ |
| Metashift | CLIP:ViT-L-14 | 32 | 20 | SGD, 0.1 | 50 | $\{5.0, 2.0, 1.0, 0.1\}$ |
| Camelyon17 | BiomedCLIP | 64 | 30 | SGD, 0.01 | 20 | $\{0.5, 1.0, 0.5, 1.0\}$ |

Table 4: Summary of the hyper-parameters used in our experiments.

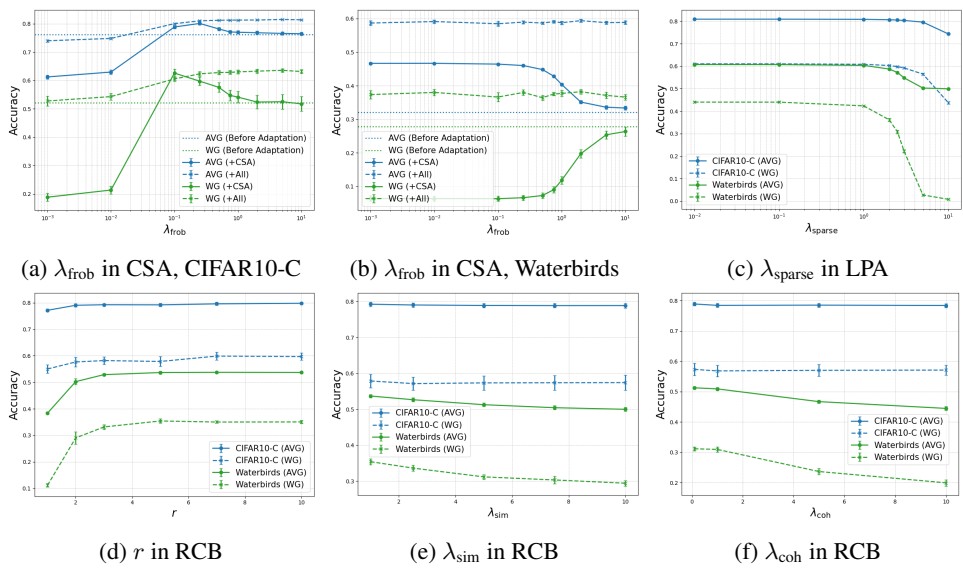

(a) $\lambda_{\text{frob}}$ in CSA, CIFAR10-C  (b) $\lambda_{\text{frob}}$ in CSA, Waterbirds  (c) $\lambda_{\text{sparse}}$ in LPA

(d) $r$ in RCB  (e) $\lambda_{\text{sim}}$ in RCB  (f) $\lambda_{\text{coh}}$ in RCB

Figure 5: **Ablations on the hyper-parameters in CONDA.** We ablate on the individual hyper-parameters in CONDA for each type of distribution shift: (1) CIFAR10-C (impulse noise) simulating low-level shift, and (2) Waterbirds simulating concept-level shift.

# D   ABLATION EXPERIMENTS

**Ablation Study on Hyperparameters.**

In Figure 5, we present a comprehensive ablation study illustrating how different hyperparameter choices affect the performance of our proposed method.

Most notably, in Figures 5a and 5b, we observe that $\lambda_{\text{frob}}$ influences the adaptation performance differently depending on the type of distribution shift (*i.e.*, CIFAR10-C for low-level shifts and Waterbirds for concept-level shifts). Recall that $\lambda_{\text{frob}}$ controls how much the concept vectors are allowed to deviate from their original construction during adaptation. When $\lambda_{\text{frob}}$ is very low (*e.g.*, 0.001), the weights in the concept bottleneck layer deviate excessively, leading to instability.

In the case of low-level shifts, as shown in Figure 5a, over-regularizing the Frobenius norm term (*e.g.*, setting $\lambda_{\text{frob}}$ as high as 10) prevents the method from addressing the non-robustness of the concept bottleneck under such shifts (*i.e.*, the first failure mode in Section 2.3). Selecting a suitable moderate value such as $\lambda_{\text{frob}} = 0.1$ leads to optimal performance.

In contrast, under concept-level shifts depicted in Figure 5b, allowing deviation of the concept vectors harms performance. By strongly regularizing with a high $\lambda_{\text{frob}}$ value (*e.g.*, $\lambda_{\text{frob}} = 10$), we can nearly preserve the original pre-adaptation performance (note that WG drops to almost zero when $\lambda_{\text{frob}} < 1$). This occurs because failures of CBMs under concept shifts need to be addressed by adapting the linear probing layer rather than the concept bottleneck layer (the second failure mode in Section 2.3).

However, when all components of CONDA are activated, the adaptation performance becomes quite insensitive to the choice of $\lambda_{\text{frob}}$, regardless of the type of distribution shift, since all the components collaboratively combine to handle all possible failure modes.

Regarding the hyperparameters for the other regularization terms – namely, $\lambda_{sparse}$, $\lambda_{sim}$, and $\lambda_{coh}$ (Figures 5c, 5e, and 5f) – we find that the performance is relatively insensitive to their values unless they are set too high, which could override the main optimization objectives.

As for the number of residual concepts $r$, shown in Figure 5d, we observe that increasing $r$ helps improve the performance up to a certain point (specifically, $r = 5$), after which the performance saturates and additional concepts become redundant. We recommend that a criterion like this be used to select $r$ in practice. Choose a high cosine similarity threshold (e.g., 0.9), and stop adding new residual concepts once a new concept vector starts to have maximum cosine similarity (with the existing set of residual concept vectors) larger than the set threshold.

**Pseudo-labeling variants.** We acknowledge that the performance of CONDA relies on the quality of pseudo-labels. In Section 3, we introduced a simple yet effective approach that ensembles the foundation model's zero-shot predictions and linear probing predictions, with a focus on matching the CBMs' post-deployment performance with that of the feature-based predictions. However, more advanced pseudo-labeling techniques could further improve our method's performance.

Importantly, the pseudo-labeling technique should operate on a batch basis to run alongside CONDA, which performs adaptation with an incoming batch of test data in an online fashion. For this, we employ a recent method from the TTA literature Chen et al. (2022). They employ an online pseudo-labeling refinement scheme that generates significantly more accurate pseudo-labels by using soft $k$-nearest neighbors voting in the target domain's feature space for each target sample. The neighboring samples are generated by applying weak augmentation to each incoming target sample. The core intuition of their method is that the model should make consistent predictions for these nearest neighbors. We apply this method to the feature-based linear-probing predictions ("Refined LP") and ensemble it with ZS predictions.

Table 5 compares the performance of CONDA with Post-hoc CBM (Yuksekgonul et al., 2023) when using i) our simple pseudo-labeling approach, ii) pseudo-label refinement by Chen et al. (2022), and iii) perfect pseudo labeling (using the ground-truth labels of the target dataset to provide an empirical upper bound on the performance). We observe that the refined pseudo-labeling of Chen et al. (2022) helps further improve the adaptation performance of CONDA. It is particularly effective with low-level shifts (CIFAR10-C and CIFAR100-C), as the method by Chen et al. (2022) enforces consistent predictions among weakly-augmented instances, which correspond to low-level shifts (*e.g.*, cropping, color jittering, flipping, *etc.*). However, compared to the performance with perfect pseudo-labeling, there remains a significant performance gap (especially CIFAR-100). Reducing this gap with more advanced pseudo-labeling that can handle both distribution shift types is an important direction for future work.

| Dataset | Metric | {ZS, LP} | {ZS, Refined LP} | Perfect PL |
|---|---|---|---|---|
| CIFAR10-C | AVG | $84.38 \pm 1.52$ | $90.06 \pm 1.94$ | $96.37 \pm 0.37$ |
| | WG | $72.69 \pm 2.49$ | $76.31 \pm 3.01$ | $92.65 \pm 0.56$ |
| CIFAR100-C | AVG | $53.88 \pm 0.23$ | $61.25 \pm 0.29$ | $97.31 \pm 0.35$ |
| | WG | $2.56 \pm 0.27$ | $10.28 \pm 0.27$ | $79.13 \pm 1.73$ |
| Waterbirds | AVG | $60.69 \pm 0.23$ | $62.77 \pm 0.16$ | $95.39 \pm 0.21$ |
| | WG | $43.01 \pm 0.46$ | $44.30 \pm 0.11$ | $92.02 \pm 0.42$ |
| Metashift | AVG | $93.69 \pm 0.20$ | $94.07 \pm 0.11$ | $100.0$ |
| | WG | $92.02 \pm 0.12$ | $93.56 \pm 0.13$ | $100.0$ |
| Camelyon17 | AVG | $91.20 \pm 0.06$ | $93.19 \pm 0.10$ | $94.82 \pm 0.08$ |
| | WG | $88.96 \pm 0.16$ | $90.88 \pm 0.15$ | $93.50 \pm 0.17$ |

Table 5: **Performance of CONDA with different pseudo-labeling techniques**. Here, ZS and LP refer to zero-shot and linear probing methods used for prediction based on the foundation model. Refined LP refers to the pseudo-labeling method of Chen et al. (2022).

**Choice of foundation model.** Another factor that inherently affects the performance of CONDA is the choice of the backbone foundation model. While foundation models are usually designed for general-purpose tasks (*e.g.*, BiomedCLIP (Zhang et al., 2023), pretrained on diverse medical domains), they are sometimes fine-tuned for specific domains (*e.g.*, MedCLIP (Wang et al., 2022), specifically pretrained on chest X-rays).

In Table 6, we compare the performance of our proposed adaptation with different CBM baselines on the Camelyon17 dataset, while varying the backbone foundation model. Intuitively, MedCLIP may not be well-suited for pathology data such as Camelyon17, and we observe significant drops in zero-shot (ZS) and linear probing (LP) accuracies in both source and target domains. Consequently, the performance of the CBM based on its embeddings is much worse when a mis-matched foundation model is used. On the other hand, with BioMedCLIP as the foundation model, the source domain performance as well as the adaptation performance of CONDA on the target domain are much better. This confirms that selecting an appropriate backbone leads to better representative embeddings and higher-quality pseudo labels, which in-turn leads to more accurate test-time adaptation.

This suggests another avenue for further improving the adaptation performance beyond advanced pseudo-labeling techniques – for example, zero-shot robustification of the foundation model embeddings (Adila et al., 2024). Such approaches could be employed when the chosen foundation model is not specifically tailored to the given task. We leave this as an important direction for future work.

| Backbone | | | ZS | LP | Yuksekgonul et al. (2023) | | Yeh et al. (2020) | | Oikarinen et al. (2023) | |
|---|---|---|---|---|---|---|---|---|---|---|
| | | | | | Unadapted | w/ CONDA | Unadapted | w/ CONDA | Unadapted | w/ CONDA |
| BiomedCLIP | Source | AVG | 77.71 | $92.14 \pm 0.01$ | $89.07 \pm 0.60$ | - | $97.01 \pm 0.05$ | - | $94.19 \pm 0.11$ | - |
| | | WG | 69.73 | $88.89 \pm 0.02$ | $84.34 \pm 1.39$ | - | $96.31 \pm 0.24$ | - | $91.23 \pm 0.12$ | - |
| | Target | AVG | 84.55 | $93.69 \pm 0.01$ | $89.71 \pm 0.65$ | $91.20 \pm 0.06$ | $95.01 \pm 0.07$ | $92.54 \pm 0.16$ | $91.75 \pm 0.08$ | $93.16 \pm 0.05$ |
| | | WG | 76.08 | $89.49 \pm 0.02$ | $85.96 \pm 0.88$ | $88.96 \pm 0.16$ | $93.07 \pm 0.37$ | $91.07 \pm 0.32$ | $87.24 \pm 0.09$ | $89.00 \pm 0.07$ |
| MedCLIP | Source | AVG | 53.09 | $79.89 \pm 0.05$ | $76.92 \pm 0.06$ | - | $94.58 \pm 0.10$ | - | $79.15 \pm 0.08$ | - |
| | | WG | 11.75 | $79.28 \pm 0.01$ | $76.21 \pm 0.16$ | - | $92.20 \pm 0.44$ | - | $78.01 \pm 0.15$ | - |
| | Target | AVG | 48.87 | $68.37 \pm 0.07$ | $67.35 \pm 0.12$ | $67.56 \pm 0.11$ | $88.72 \pm 0.28$ | $86.04 \pm 0.19$ | $66.29 \pm 0.18$ | $67.05 \pm 0.08$ |
| | | WG | 14.66 | $68.32 \pm 0.05$ | $62.15 \pm 0.19$ | $65.36 \pm 0.14$ | $81.42 \pm 1.15$ | $81.01 \pm 1.65$ | $59.35 \pm 0.21$ | $65.17 \pm 0.14$ |

Table 6: **Performance of CONDA varying backbone foundation model.** The dataset is Camelyon17, simulating a natural shift between the source and target domains.

# E    ADDITIONAL INTERPRETABILITY ANALYSIS

In this section, we include additional experiments and analysis to better understand the interpretability of CONDA as well as the utility of the RCB component.

## E.1    RESIDUAL CONCEPT BOTTLENECK COMPENSATES FOR PREDICTION ERRORS

Here we aim to understand how including the RCB component in CONDA impacts the predictions of the adapted classifier. We conduct an analysis similar to the one in Appendix B of Yuksekgonul et al. (2023), where they evaluate the impact of the residual predictor PCBM-h and when it alters the predictions of the main predictor PCBM. In Figure 6, we compare the predictions made by (i) PCBM + CSA + LPA (*i.e.*, excluding RCB) with that of (ii) PCBM + CSA + LPA + RCB (*i.e.*, including RCB) on the CIFAR10-C (with Gaussian Noise, Shot Noise, and Impulse Noise) and Metashift datasets. The x-axis shows the confidence of predictions, which are binned into 5 intervals; and y-axis shows both the accuracy of (i) within each confidence bin (blue curve), and the consistency of predictions between (i) and (ii) within each confidence bin (red curve). Consistency is defined as the fraction of samples where the predictions of two models are the same.

Figures 6a and 6b show the accuracy/consistency plots for the CIFAR10-C and Metashift datasets respectively. We observe that in both cases, when the confidence is high, the accuracy and consistency are high. As the confidence of predictions decreases, the accuracy and consistency within the confidence bins also decrease sharply. From this, we can infer that the residual component (RCB) modifies the predictions of PCBM + CSA + LPA mostly when they are incorrect and have low confidence. This is readily apparent in the case of Metashift which addresses binary classification, since all the inconsistent predictions where PCBM + CSA + LPA is incorrect have to be correct when RCB is included. Thus, we hypothesize that RCB has the effect of intervening to compensate mainly when the prior adaptation components (CSA + LPA) have prediction errors or low confidence.

We also summarize the test accuracies of the CONDA variants (i) and (ii) on Metashift and CIFAR10-C in Table 7. We observe that including RCB (variant ii) leads to a small increase in both the AVG and WG accuracies.

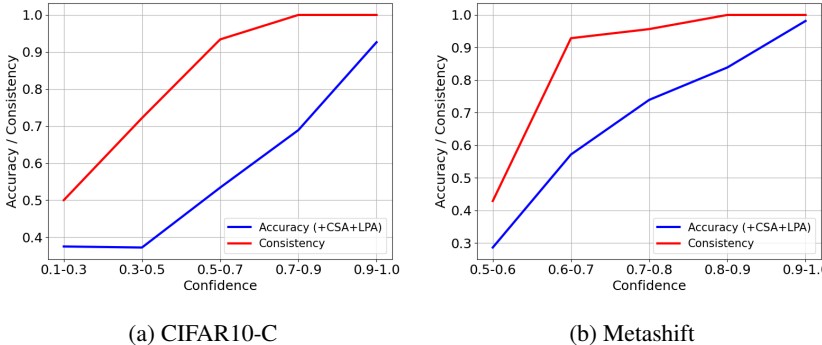

(a) CIFAR10-C  (b) Metashift

Figure 6: RCB intervenes and compensates for mis-classifications of PCBM + CSA + LPA, particularly in the lower confidence prediction bands. For Metashift, with two classes, a mis-classification and inconsistent prediction imply that RCB corrects the prediction.

| Dataset | CONDA variant | Accuracy (AVG) | Accuracy (WG) |
|---|---|---|---|
| Metashift | PCBM + CSA + LPA | 93.96 | 91.61 |
| | PCBM + CSA + LPA + RCB | 94.57 | 93.25 |
| CIFAR10-C | PCBM + CSA + LPA | 81.89 | 64.63 |
| | PCBM + CSA + LPA + RCB | 82.27 | 66.54 |

Table 7: Accuracy comparison of the CONDA variants where (i) RCB is excluded and (ii) RCB is included on Metashift and CIFAR10-C (with Gaussian Noise, Shot Noise, and Impulse Noise).

### E.2 ACCURACY-INTERPRETABILITY TRADEOFF IN RESIDUAL CONCEPT BOTTLENECK

In this sub-section, we aim to answer to the following question: while the residual concept bottleneck improves the adaptability of CONDA, does it potentially affect the interpretability by introducing additional model complexity? An analysis of the trade-off between model complexity and interpretability, particularly as new residual concepts are added, would be valuable for practitioners seeking interpretable yet robust models.

Here we apply our adaptation to PCBM (CLIP), where each concept vector is constructed using CLIP text embeddings of concept captions, deployed to the Waterbirds dataset. As discussed in Appendix B.2, for concept annotation, we leveraged the ConceptNet hierarchy following the setup in Yuksekgonul et al. (2023). We searched ConceptNet for the words "Bird", "Water", "Land" and obtained concepts that have the following relationship with the query concept: `hasA`, `isA`, `partOf`, `HasProperty`, and `MadeOf`.

We compare the two CONDA variants (i) PCBM + CSA + LPA and (ii) PCBM + CSA + LPA + RCB by varying the number of residual concepts ($r$) and evaluating the following metrics:

– Relative accuracy of method (ii) minus (i), both for AVG and WG.
– Similarity score output in Eqn. 18 from the automatic concept annotation method described in Appendix B.2.

The similarity score is used as a quantitative metric to measure the interpretability of RCB. To be more specific, the score in Eqn. 18 tells us how aligned the assigned concept caption is with each residual concept vector. A low score implies that the assigned caption is not a good description for the concept.

Figure 7 shows the relative accuracy and similarity score as a function of the number of residual concepts (as they are added incrementally). We observe that choosing $r = 5$ would result in the best relative accuracy, but it leads to a drop in the similarity score which peaks at $r = 4$ (implying a drop in interpretability of the residual concept). A practitioner can choose a suitable stopping point for the residual concepts by monitoring these two criteria as shown in the figure.

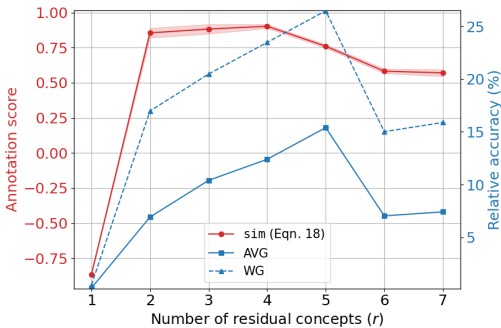

Figure 7: **Accuracy vs. Interpretability of the Residual Concept Bottleneck.** with PCBM (CLIP) deployed on the Waterbirds dataset (target domain).

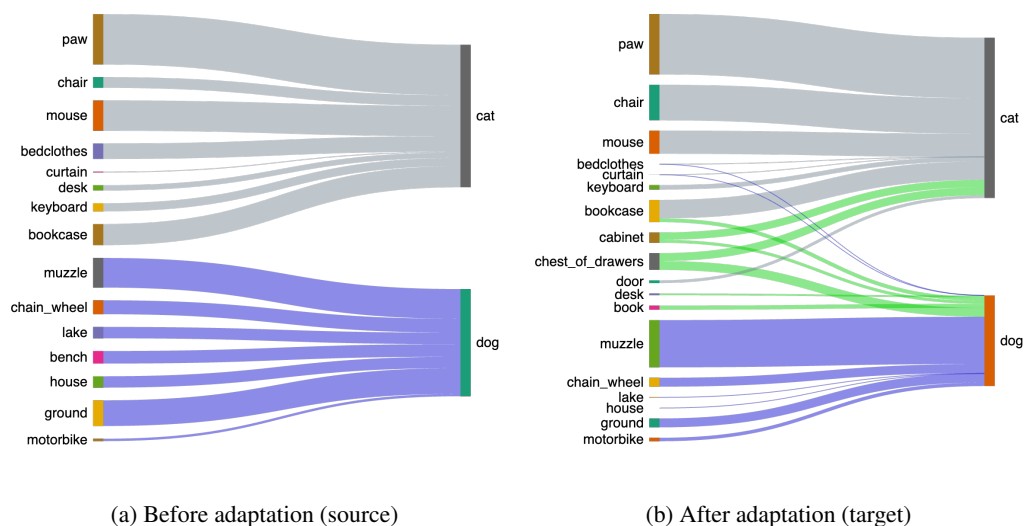

(a) Before adaptation (source)  (b) After adaptation (target)

Figure 8: **CONDA adapts the concept weights to be tailored to the target data.** We visualize the linear probing layer weights (width of each mapping) before vs after applying CONDA to PCBM (Yuksekgonul et al., 2023) on the Metashift dataset. We only show the mappings with positive weights.

### E.3    ADDITIONAL INTERPRETABILITY RESULTS

In Figure 8, we present another example demonstrating how CONDA adapts interpretations on the MetaShift dataset, similar to Figure 4. In the source domain, cat images are exclusively correlated with sofa or bed objects, whereas dog images are always associated with bench or bike objects. In the target domain, however, both cat and dog images appear with a shelf background.

Without any adaptation, the deployed CBM indicates that the most contributing concepts to the "cat" class are mainly household-related objects (see Figure 8a), and these concepts do not positively contribute to the "dog" class at all. After applying our adaptation (Figure 8b), the influence of the bed-related concepts is diminished, while shelf-related concepts (highlighted in bright green) begin to contribute to the prediction of both the "cat" and "dog" classes.

## F    LIMITATIONS AND FUTURE WORK

This work was motivated by our observation that recent CBM variants atop a backbone foundation model may close the performance gap with feature-based predictions in the source domain, but they

are often unable to do so under distribution shifts at test time (after deployment). Hence, for an interpretable and robust decision-making pipeline under distribution shifts, while fully leveraging the representative power of foundation models, an adaptive test-time approach is required. To the best of our knowledge, we have proposed the first effort to tackle this problem setting for CBMs. We formalized potential failure modes under low-level and concept-level distribution shifts and proposed a novel test-time adaptation framework, named CONDA. Each component of CONDA is designed to address specific failure modes, effectively improving the test-time performance of a deployed CBM using *only unlabeled* test data.

We acknowledge that the effectiveness of our framework is limited by the inherent robustness of the backbone foundation model, especially due to its reliance on pseudo-labeling. Specifically, when the backbone foundation model remains robust (*e.g.*, against low-level shifts with lower severity level or concept-level shifts), concept-based predictions can be adjusted to be more robust than feature-based predictions through adaptation (*e.g.*, see Metashift results in Table 1). However, when the backbone foundation model is *not robust* (*e.g.*, against low-level shifts with higher severity level), the CBM adaptation, which relies on the pseudo-labels of the foundation model (feature representations), cannot be guided to a successful solution and could lead to reduced performance; see results in Table 8. Moreover, in Table 1, we note that there are instances where our adaptation did not yield

| Dataset | | ZS | LP | Yuksekgonul et al. (2023) | | | | Yeh et al. (2020) | | | |
|---|---|---|---|---|---|---|---|---|---|---|---|
| | | | | w/o adaptation | + CSA | + LPA | + CSA + LPA | w/o adaptation | + CSA | + LPA | + CSA + LPA |
| Metashift | Source AVG | 0.957 | 0.972 | 0.979 ± 0.001 | - | - | - | 0.972 ± 0.001 | - | - | - |
| | Source WG | 0.934 | 0.960 | 0.969 ± 0.003 | - | - | - | 0.960 ± 0.001 | - | - | - |
| | Target AVG | 0.705 | 0.835 | 0.890 ± 0.006 | 0.620 ± 0.049 | 0.713 ± 0.005 | 0.676 ± 0.009 | 0.840 ± 0.009 | 0.834 ± 0.009 | 0.749 ± 0.008 | 0.690 ± 0.005 |
| | Target WG | 0.460 | 0.720 | 0.850 ± 0.013 | 0.279 ± 0.110 | 0.476 ± 0.017 | 0.398 ± 0.018 | 0.712 ± 0.018 | 0.700 ± 0.020 | 0.512 ± 0.016 | 0.400 ± 0.010 |

Table 8: **Negative results of our test-time adaptation.** In the target domain, the model faces Metashift images with random Gaussian noise (severity level five), following the implementation of Hendrycks & Dietterich (2019). When the performance of zero-shot and linear-probing inference is poor on the target domain, the pseudo-labels cannot serve as a reliable reference for the test-time adaptation. Therefore, the performance of CONDA with different components on the target domain is worse than that of the model without any adaptation.

improvements with the CBM method of Yeh et al. (2020). In cases such as the CIFAR datasets and Camelyon17, the unadapted CBM already outperforms ZS or LP in the target domain, and adaptation using pseudo-labels produced by these methods can negatively impact the performance. This is likely because the concept learning algorithm in Yeh et al. (2020) is designed to optimize accuracy, with the concept bottleneck layer serving as an additional layer that can be optimized along with the subsequent LP layer. However, a caveat of this approach is that the interpretability of the concept bottleneck is not guaranteed, whereas methods such as Yuksekgonul et al. (2023) and Oikarinen et al. (2023) provide clear textual annotations for concepts, enhancing the interpretability.

Despite these limitations, we believe our work is an important first step toward leveraging off-the-shelf foundation models in an interpretable decision-making process, while preserving the post-deployment utility. We highlight that our framework can continue to benefit from ongoing improvements in the robustness of foundation models and the development of more advanced pseudo-labeling techniques (as hinted in Appendix D), both of which represent promising avenues for future work. Another promising direction for future research is to develop a deeper theoretical understanding of concept bottlenecks under distribution shifts. For instance, it would be valuable to i) characterize the sufficiency of a given concept set from training (source domain) for robust test-time accuracy under different distribution shifts; and ii) to quantify or bound the extent to which test-time adaptation can bridge the accuracy gap between the source and target distributions. Such theoretical insights would complement the algorithmic and empirical advancements, guiding both the design of more effective residual concept bottleneck and the development of improved adaptation strategies.

