# OpenReview forum: "CONDA: Adaptive Concept Bottleneck for Foundation Models Under Distribution Shifts"
_ICLR.cc/2025/Conference — ICLR 2025 Poster_

### Official Review · Reviewer_yYWh · 2024-10-30

**Soundness:** 2
**Presentation:** 2
**Contribution:** 2
**Rating:** 6
**Confidence:** 3

**Summary:**

This paper studies an interesting problem, i.e., how to transform non-interpretable foundation models into concept-based interpretable decision-making pipelines under different types of distribution shifts. To address this problem, this paper proposes an adaptive concept bottleneck framework (CONDA) to address these distribution shifts, that dynamically adapts the concept-vector bank and the prediction layer based on unlabeled data from the target domain. The authors also conduct experiments to evaluate the performance of the proposed method.

**Strengths:**

[+] This paper lists several possible failure modes of the decision-making pipeline of a foundation model equipped with a Concept Bottleneck Model (CBM)

[+] To handle the potential incomplete and new concepts to bridge the distribution gap between the source and target domains, this paper introduces a residual CBM with additional concept vectors and a linear predictor.

**Weaknesses:**

[-] More explanations should be provided to clarify the rationale behind the techniques adopted in the proposed method. For example, this paper introduces $r$ additional concept vectors. Could you clarify the potential criteria to select these additional vectors? What are the connections between these different criteria? How would different criteria influence the performance of your proposed method?

[-] The authors fail to provide the complexity analysis of the proposed method. The proposed method consists of some components, which could involve several different steps (e.g., using an ensemble of zero-shot predictor and linear probing predictor to get pseudo-labels). It would be better if the authors could provide the detailed complexity analysis of the proposed method.

[-] The experiments to support the proposed method are not sufficient. For example, it is unclear about how the cosine similarity-based regularization in the objective can ensure that the new concept vectors are "minimally" redundant with each other and have "minimal"  overlap with existing concept vectors? It would be helpful if they could include experiments and a high-level analysis to support this "minimal" overlap.

[-] Some typos, e.g., Appendix ?? in line 875.

**Questions:**

[1] This paper introduces $r$ additional concept vectors. Could you clarify the potential criteria to select these additional vectors? What are the connections between these different criteria? How would different criteria influence the performance of your proposed method?

[2] This paper mentions that the parameters of the residual CBM are randomly initialized. What are the potential effective randomization techniques for the residual CBM? Do these different randomization techniques have difference influence degrees on the performance of the proposed method?

[3] How to show that the adopted cosine similarity-based regularization in the objective can ensure that the new concept vectors in are minimally redundant with each other, and also have minimal overlap with the existing concept vectors? It would be better if the authors could provide the experiments and high-level analysis for this.

[4] How to determine the effective top-$k$ nearest neighbors for different $\tilde{c}_{i}$? How would different choices influence of the performance of the proposed method?

---

### Official Review · Reviewer_wVbS · 2024-11-01

**Soundness:** 2
**Presentation:** 3
**Contribution:** 2
**Rating:** 5
**Confidence:** 4

**Summary:**

The paper introduces CONDA, a framework that dynamically adapts Concept Bottleneck Models (CBMs) to handle distribution shifts using only unlabeled target domain data, thereby improving interpretability and performance.

**Strengths:**

The research is well-designed and comprehensive, with robust experimental results across multiple datasets, demonstrating significant improvements in accuracy and interpretability. The paper is well-written and clear, making the technical aspects and findings accessible. Its significance lies in enhancing the robustness and interpretability of foundation models in real-world applications, particularly in safety-critical domains where distribution shifts are common.

**Weaknesses:**

1) The paper does not thoroughly explore the sensitivity of the results to different hyperparameters.
2) The computational efficiency of CONDA is not extensively discussed.  Evaluating the runtime and resource requirements, especially for large-scale datasets, would be valuable for practical applications.
3) The paper focuses on accuracy improvements but could benefit from more quantitative metrics to evaluate the interpretability of the adapted models.  This would provide a more comprehensive assessment of the framework's impact on interpretability. I'm actually a researcher in this area, and I've been wondering about interpretable metrics for this kind of unsupervised/unlabled CBM task, so I didn't mean to be hard on you, I just wanted to hear what you have to say about the evaluation of interpretable credibility in this area.

**Questions:**

1) Could you explore more advanced pseudo-labeling methods, such as weak and strong augmentations or soft nearest-neighbor voting, to see if they improve the robustness and accuracy of the adaptation process?
2) Besides accuracy, what quantitative metrics did you use to evaluate the interpretability of the adapted models?  Could you provide more detailed results on interpretability?
3) Have you tested CONDA with different types of foundation models and concept bottleneck constructions?  If so, what were the results, and if not, what are your thoughts on its generalizability?
4) Could you perform a sensitivity analysis to understand how different hyperparameters affect the performance of CONDA, and provide guidelines for tuning these parameters?
5) It makes me feel that the overall workload/experiment/contribution is limited. In addition, I would also like to see some case studies and visualizations to better understand your problems and methods. Overall, I find this paper a bit like an unfinished version, but I have to admit that the problem setting is actually quite interesting and meaningful.

I look forward to talking to the authors during rebuttal to get a better understanding of the paper. My initial score is only based on my current understanding of this paper. I really hope that I can better understand the whole paper and improve my score during the rebuttal.

---

### Official Review · Reviewer_VnZY · 2024-11-02

**Soundness:** 3
**Presentation:** 3
**Contribution:** 3
**Rating:** 5
**Confidence:** 4

**Summary:**

This paper investigates the potential of transforming complex and non-interpretable Foundation Models (FMs) into interpretable models using Concept Bottleneck Models (CBMs). Specifically, it focuses on building robust models that maintain strong performance under distribution shifts through test-time adaptation. The authors categorize three types of failure modes where CBMs may struggle under distribution shifts—low-level shift, concept-level shift, and incomplete concept set—and propose a framework called CONDA to address each. CONDA comprises three modules: Concept-Score Alignment, Linear Probing Adaptation, and Residual Concept Bottleneck. Experimental results demonstrate that these modules effectively mitigate failure modes, thereby enhancing model robustness and interpretability in various challenging scenarios.

**Strengths:**

•  This paper addresses a significant gap by focusing on test-time adaptation for Concept Bottleneck Models (CBMs), a rarely explored area. By tackling the robustness of interpretable models under distribution shifts, the paper contributes valuable insights into making foundational model-based pipelines more practical and trustworthy in real-world scenarios.

• The proposed CONDA framework is well-structured with three distinct modules—Concept-Score Alignment, Linear Probing Adaptation, and Residual Concept Bottleneck. Each module addresses a specific type of distribution shift failure mode, allowing for a comprehensive approach to adapting CBMs dynamically without compromising interpretability.

•  The authors conduct experiments across a diverse range of datasets, including CIFAR, Waterbirds, and Camelyon17, to validate CONDA’s effectiveness under various distribution shifts. This diversity strengthens the claim that CONDA generalizes well across different domains and types of shifts, reinforcing its potential applicability to real-world challenges.

•  The paper demonstrates up to a 28% improvement in test-time accuracy over standard CBM approaches, which is a substantial gain. This improvement, particularly in challenging distribution-shifted settings, highlights the framework’s robustness and establishes its effectiveness in bridging the performance gap between interpretable and non-interpretable models.

**Weaknesses:**

•  Lack of Depth in the Related Work Section:
The related work section could be expanded to enhance readability and provide a more comprehensive overview of relevant literature. It would be beneficial to include a discussion of label-free CBMs and related methods. The authors should consider incorporating additional references to enrich the context, particularly those exploring label-free CBMs.

(1) Oikarinen, T., Das, S., Nguyen, L. M., & Weng, T. W. Label-free Concept Bottleneck Models. In The Eleventh International Conference on Learning Representations.

(2) Wang, B., Li, L., Nakashima, Y., & Nagahara, H. (2023). Learning bottleneck concepts in image classification. In Proceedings of the ieee/cvf conference on computer vision and pattern recognition (pp. 10962-10971).

(3) Shang, C., Zhou, S., Zhang, H., Ni, X., Yang, Y., & Wang, Y. (2024). Incremental residual concept bottleneck models. In Proceedings of the IEEE/CVF Conference on Computer Vision and Pattern Recognition (pp. 11030-11040).


•  Unclear Articulation of Contributions in the Introduction:
The introduction section lacks clarity regarding the specific contributions of the paper. The authors should refine this section to explicitly outline their contributions to improve the reader’s understanding of the paper’s unique impact.

•  Need for More Comparative Experiments with Baseline CBMs:
The experimental results could be strengthened by adding comparisons to traditional CBM models. Including analyses of concept and task accuracy would also provide valuable insights into the framework's improvements over standard CBMs.

•  Limited Dataset Diversity in Experiments:
The study would benefit from additional experiments on commonly used CBM datasets such as AwA2, CelebA, CUB (Caltech-UCSD Birds) and TravelingBirds. Expanding to these datasets, along with comparisons to other state-of-the-art CBM models, could further validate the generalizability and competitiveness of the proposed method.

•  Reference Error on Line 875:
There is a referencing error on line 875, which should be corrected to improve the document's accuracy and professionalism.

•  Reliance on Foundation Model Robustness Assumptions:
The pseudo labeing effectiveness assumes that the feature extraction from foundation models remains robust under distribution shifts, which might not hold in all real-world scenarios. Evaluating the performance with varied foundation models or explicitly testing robustness assumptions could help clarify these dependencies.

•  Limited Analysis of Interpretability-Complexity Trade-off for Residual Concepts:
While the residual concept bottleneck improves adaptability, it potentially introduces complexity that may affect interpretability. An analysis of the trade-off between model complexity and interpretability, particularly as new residual concepts are added, would be valuable for practitioners seeking interpretable yet robust models.

**Questions:**

see weaknesses

---

### Official Review · Reviewer_uv21 · 2024-11-04

**Soundness:** 3
**Presentation:** 3
**Contribution:** 2
**Rating:** 6
**Confidence:** 4

**Summary:**

This paper investigates an adaptive concept bottleneck approach for converting non-interpretable foundation models into interpretable decision-making pipelines using high-level concept vectors. This method is particularly useful for handling distribution shifts, especially when there is unlabeled data from the target domain. The key ideas behind this work are inspired by Concept Bottleneck Models (CBM) and Test-Time Adaptation (TTA).

This paper provides several insightful observations: (1) a naive application of CBMs is insufficient for fully leveraging the robustness and expressiveness of foundation model (FM) features under test-time shifts, as illustrated in Figure 1; (2) the identification of failure modes in concept bottlenecks for foundation models, based on a definition of distribution shifts “in the wild.” Based on these observations, the paper proposes a three-stage approach to align the concept bottleneck, label predictor, and Residual Concept Bottleneck, allowing for the extension of additional concept vectors. This approach is combined with a new architectural design and regularization strategies to enhance test-time adaptation, coherency, and interpretability.

Experimental testing on five datasets and three baselines demonstrates that the proposed approach effectively improves the test-time performance of deployed CBMs in most cases.

**Strengths:**

1. This work is the first to study the post-deployment performance of concept bottlenecks for foundation models. The research question is both interesting and practical.

2. The experimental results are reasonable, demonstrating that the proposed method improves performance in most cases.

3. The writing is well-organized, clearly outlining the scope, motivation, and insights regarding shift types and failure modes of concept bottlenecks during test-time adaptation.

**Weaknesses:**

1. The definition of a Concept Bottleneck should be clarified. From my reading of the paper, it seems that the concept bottleneck is a learnable mapping function that transforms hidden features from foundation models into lower-dimensional concept features. However, it’s unclear whether this bottleneck acts as a dictionary or codebook that links concepts (text) to embeddings. Additionally, a more thorough introduction to the three baselines used for constructing the concept bottleneck is needed. The current explanations are somewhat abstract and too brief, making it difficult for readers to grasp the details.

2. The ablation study reveals that the influence of individual components is inconsistent. In different scenarios, different components demonstrate varying levels of importance. In some cases, individual components even outperform hybrid strategies, or all components have a negative influence in the worst cases. (Additionally, please correct the left plot in Figure 3.) Rather than focusing solely on individual contributions, it would be more informative to evaluate combinations of components. Specifically, removing components one by one to assess the impact on performance using the remaining components would provide a clearer understanding of their collective contributions.

3. The poor performance on the Camelyon17 dataset is largely due to the inappropriate choice of foundation model. MedCLIP, which is pretrained on chest X-rays, is not well-suited for pathology data such as Camelyon17. To fairly demonstrate the effectiveness of the proposed method, the authors should use data from the chest X-ray domain, as the foundation model significantly impacts pseudo-labeling quality. Alternatively, models like BiomedGPT (Zhang, Kai, et al. “A generalist vision–language foundation model for diverse biomedical tasks.” Nature Medicine (2024): 1-13) or BiomedCLIP (Zhang, Sheng, et al. “BiomedCLIP: a multimodal biomedical foundation model pretrained from fifteen million scientific image-text pairs.” arXiv preprint arXiv:2303.00915 (2023)) could be more appropriate due to their pretraining on diverse medical domains. Additionally, the results highlight the significant influence of foundation models on performance. From my perspective, this is a crucial analysis point that should be explored in more depth in this paper.

**Questions:**

The main question concerns the definition of the concept bottleneck, as outlined in the “weakness” section. I suggest that the authors provide a concrete explanation or illustrative figures to clarify this concept within the context of the paper.

My primary concern regarding the acceptance of this paper is the necessity of introducing the residual concept bottleneck, as it does not appear to offer significant benefits in modeling. Although a case study shows adjustments in the concept-to-class mappings (Figure 4), this is based on a single example. I recommend providing a more comprehensive analysis, ideally with quantified interpretation results, to better explain this phenomenon.

---

### Meta-Review · Area_Chair_4HgM · 2024-12-18

**Metareview:**

The paper introduces a novel framework aimed at enhancing the robustness and interpretability of Concept Bottleneck Models (CBMs) under distribution shifts. One of its key strengths lies in the first effort to explore Test-Time Adaptation (TTA) for CBMs in conjunction with foundation models. By addressing key failure modes associated with distribution shifts—namely low-level shifts, concept-level shifts, and incomplete concept sets—the authors provide a clear problem formulation.

A significant contribution of the paper is the introduction of three adaptive components: Concept-Score Alignment (CSA), Linear Probing Adaptation (LPA), and Residual Concept Bottleneck (RCB). Each component addresses specific failure modes, thereby enabling the model to maintain robust performance under diverse shift scenarios. The RCB helps to introduce new concepts not present in the original concept set, hence enhancing the interpretability and robustness of the model. The authors conduct experiments on multiple datasets, including CIFAR-10, CIFAR-100, Waterbirds, Metashift, and Camelyon17, covering a range of distribution shifts (low-level, concept-level, and natural shifts). These experiments show that CONDA can improve Average Group Accuracy (AVG) and Worst Group Accuracy (WG), with improvements of up to 28% in some cases. Additionally, the paper emphasises interpretability by showing how the introduction of residual concepts, like "feathers" and "wings" in Waterbirds, aligns with human intuition.

However, the paper has some clear weaknesses. One of the main challenges is its reliance on the assumption of  "concept set completeness", which may not always hold in real-world scenarios. While the RCB attempts to address this by introducing new concepts, the process for ensuring that all relevant concepts are captured remains somewhat opaque. Another potential limitation is the dependence on pseudo-labels for adaptation. Since pseudo-labels are inferred from the unadapted model, they may introduce noise, especially in cases of severe distribution shifts. While the authors mitigate this issue through ensemble-based pseudo-labeling, the impact of noisy labels on model performance is not thoroughly analyzed.

Theoretical guarantees are also missing from the paper. While the empirical results are interesting, a formal analysis of convergence or robustness guarantees under different shift conditions would have provided a stronger foundation for the proposed method. Additionally, the datasets used for evaluation remain limited. The effectiveness of CONDA on larger, more complex, or domain-specific datasets (remains an open question. Another concern is the "adaptation cost" during deployment. The need for online adaptation for each test batch may introduce computational costs, which could prevent its application in real-time systems. Finally, while the paper highlights the interpretability benefits of residual concepts, the process of discovering and explaining these concepts could be more transparent. While examples like "feathers" and "wings" are good, a deeper qualitative analysis of other discovered concepts would provide greater insight into the model's reasoning process.

In summary, the paper makes adequate advances in improving the robustness and interpretability of CBMs, especially under distribution shifts, through the proposed CONDA framework. Its strengths lie in its originality, clear problem formulation, design of adaptive components, and good empirical validation. However, reliance on pseudo-labels, adaptation costs, limited theoretical analysis, and limited benchmark datasets are somewhat missing. Despite these limitations, I think the paper is worth publishing.

**Additional Comments On Reviewer Discussion:**

Reviewers have been split in this submission, with 2 voting to accept and 2 to reject.

They all acknowledge the novelty of the system and the individual components. One concern related to the significant influence of the pre-trained backbone. While this dependency seems reasonable, it may limit the scalability and flexibility of the proposed method. However, for an interpretable and robust decision-making pipeline under distribution shifts, fully leveraging the representational power of foundation models alongside an adaptive test-time approach is essential. The reviewer highlighted that this paper is the first to explore this problem setting for CBMs, making it a valuable contribution to the field.

The main stumbling point, therefore, concerns the experimental setup, i.e. results on benchmark datasets are limited. I do not think this is necessarily the case, largely because the authors have explained: "We evaluate the performance of concept bottlenecks for FMs and the proposed adaptation on five real-world datasets with distribution shifts, following the setup in Lee et al. (2023): (1) CIFAR10 to CIFAR10-C and CIFAR100 to CIFAR100-C for low-level shift, (2) Waterbirds and Metashift for concept-level shift, and (3) Camelyon17 for natural shift."

I believe the experiments and ablation studies are adequate to prove the key claims of the paper within reason.

---

### Decision · Program_Chairs · 2025-01-22

Accept (Poster)